# Comparing distributions: $\ell_1$ geometry improves kernel two-sample testing

**Meyer Scetbon**
CREST, ENSAE & Inria, Université Paris-Saclay

**Gaël Varoquaux**
Inria, Université Paris-Saclay

## Abstract

Are two sets of observations drawn from the same distribution? This problem is a two-sample test. Kernel methods lead to many appealing properties. Indeed state-of-the-art approaches use the $L^2$ distance between kernel-based distribution representatives to derive their test statistics. Here, we show that $L^p$ distances (with $p \geq 1$) between these distribution representatives give metrics on the space of distributions that are well-behaved to detect differences between distributions as they metrize the weak convergence. Moreover, for analytic kernels, we show that the $L^1$ geometry gives improved testing power for scalable computational procedures. Specifically, we derive a finite dimensional approximation of the metric given as the $\ell_1$ norm of a vector which captures differences of expectations of analytic functions evaluated at spatial locations or frequencies (i.e, features). The features can be chosen to maximize the differences of the distributions and give interpretable indications of how they differs. Using an $\ell_1$ norm gives better detection because differences between representatives are dense as we use analytic kernels (non-zero almost everywhere). The tests are consistent, while much faster than state-of-the-art quadratic-time kernel-based tests. Experiments on artificial and real-world problems demonstrate improved power/time tradeoff than the state of the art, based on $\ell_2$ norms, and in some cases, better outright power than even the most expensive quadratic-time tests.

We consider two sample tests: testing whether two random variables are identically distributed without assumption on their distributions. This problem has many applications such as data integration [4] or automated model checking [22]. Distances between distributions underlie progress in unsupervised learning with generative adversarial networks [20, 1]. A kernel on the sample space can be used to build the Maximum Mean Discrepancy (MMD) [11, 12, 13, 26], a metric on distribution which has the strong propriety of metrizing the weak convergence of probability measures. It leads to non-parametric two-sample tests using the reproducing kernel Hilbert space (RKHS) distance [15, 9], or energy distance [32, 3]. The MMD has a quadratic computational cost, which may force to use of subsampled estimates [33, 14]. [5] approximate the $L^2$ distance between distribution representatives in the RKHS, to compute in linear time a pseudo metric over the space of distributions. Such approximations are related to random (Fourier) features, used in kernels algorithms [24, 19]. Distribution representatives can be mean embeddings [29, 30] or smooth characteristic functions [5, 17].

We first introduce the state of the art on Kernel-based two-sample testing built from the $L^2$ distance between mean embeddings in the RKHS. In fact, a wider family of distance is well suited for the two-sample problem: we show that for any $p \geq 1$, the $L^p$ distance between these distribution representatives is a metric on the space of Borel probability measures that metrizes their weak convergence. We then define our $\ell_1$-based statistic derived from the $L^1$ geometry and study its asymptotic behavior. We consider the general case where the number of samples of the two distributions may differ. We show that using the $\ell_1$ norm provides a better testing power. Indeed, test statistics approximate such metrics and are defined as the norm of a $J$-dimensional vector which is the difference between the two distribution representatives at $J$ locations. Under the alternative hypothesis $H_1$: $P \neq Q$, the

analyticity of the kernel ensures that all the features of this vector are non zero almost surely. We show that the $\ell_1$ norm captures this dense difference better than the $\ell_2$ norm and leads to better tests. We show also that improvements of Kernel two-sample tests established with the $\ell_2$ norm [17] hold in the $\ell_1$ case: optimizing features and the choice of kernel. We adapt the construction in the frequency domain as in [5]. Finally, we show that on 4 synthetic and 3 real-life problems, our new $\ell_1$-based tests outperform the state of the art.

# 1 Prior art: kernel embeddings for two-sample tests

Given two samples $X := \{x_i\}_{i=1}^n$, $Y := \{y_i\}_{i=1}^n \subset \mathcal{X}$ independently and identically distributed (i.i.d.) according to two probability measures $P$ and $Q$ on a metric space $(\mathcal{X}, d)$ respectively, the goal of a two-sample test is to decide whether $P$ is different from $Q$ on the basis of the samples. Kernel methods arise naturally in two-sample testing as they provide Euclidean norms over the space of probability measures that metrize the convergence in law. To define such a metric, we need first to introduce the notion of Integral Probability Metric (IPM):

$$\text{IPM}[F, P, Q] := \sup_{f \in F} \Big( \mathbb{E}_{x \sim P}\left[f(x)\right] - \mathbb{E}_{y \sim Q}\left[f(y)\right] \Big) \tag{1}$$

where $F$ is an arbitrary class of functions. When $F$ is the unit ball $B_k$ in the RKHS $H_k$ associated with a positive definite bounded kernel $k : \mathcal{X} \times \mathcal{X} \to \mathbb{R}$, the IPM is known as the Maximum Mean Discrepancy (MMD) [11], and it can be shown that the MMD is equal to the RKHS distance between so called mean embeddings [13],

$$\text{MMD}[P, Q] = \|\mu_P - \mu_Q\|_{H_k} \tag{2}$$

where $\mu_P$ is an embedding of the probability measure $P$ to $H_k$,

$$\mu_P(t) := \int_{\mathbb{R}^d} k(x, t) dP(x) \tag{3}$$

and $\|.\|_{H_k}$ denotes the norm in the RKHS $H_k$. Moreover for kernels said to be *characteristic* [10], *eg* Gaussian kernels, $\text{MMD}[P, Q] = 0$ if and only if $P = Q$ [11]. In addition, when the kernel is bounded, and $\mathcal{X}$ is a compact Hausdorff space, [28] show that the MMD metrizes the weak convergence. Tests between distributions can be designed using an empirical estimation of the MMD.

A drawback of the MMD is the computation cost of empirical estimates, these being the sum of two U-statistics and an empirical average, with a quadratic cost in the sample size.

[5] study a related expression defined as the $L^2$ distance between mean embeddings of Borel probability measures:

$$d_{L^2,\mu}^2(P, Q) := \int_{t \in \mathbb{R}^d} \Big|\mu_P(t) - \mu_Q(t)\Big|^2 d\Gamma(t) \tag{4}$$

where $\Gamma$ is a Borel probability measure. They estimate the integral (4) with the random variable,

$$d_{\ell_2,\mu,J}^2(P, Q) := \frac{1}{J} \sum_{j=1}^J \Big|\mu_P(T_j) - \mu_Q(T_j)\Big|^2 \tag{5}$$

where $\{T_j\}_{j=1}^J$ are sampled i.i.d. from the distribution $\Gamma$. This expression still has desirable metric-like properties, provided that the kernel is *analytic*:

**Definition 1.1** (Analytic kernel). *A positive definite kernel* $k : \mathbb{R}^d \times \mathbb{R}^d \to \mathbb{R}$ *is* analytic *on its domain if for all* $x \in \mathbb{R}^d$*, the feature map* $k(x, .)$ *is an analytic function on* $\mathbb{R}^d$*.*

Indeed, for $k$ a definite positive, characteristic, analytic, and bounded kernel on $\mathbb{R}^d$, [5] show that $d_{\ell_2,\mu,J}$ is a random metric[1] from which consistent two-sample test can be derived. By denoting $\mu_X$ and $\mu_Y$ respectively the empirical mean embeddings of $P$ and $Q$,

$$\mu_X(T) := \frac{1}{n} \sum_{i=1}^n k(x_i, T), \qquad \mu_Y(T) := \frac{1}{n} \sum_{i=1}^n k(y_i, T)$$

$$\widehat{d}_{\ell_2,\mu,J}^2[X,Y] := n \sum_{j=1}^J \left| \mu_X(T_j) - \mu_Y(T_j) \right|^2 \tag{6}$$

converges in distribution to a sum of correlated chi-squared variables. Moreover, under the alternative hypothesis $H_1 : P \neq Q$, $\widehat{d}_{\ell_2,\mu,J}^2[X,Y]$ can be arbitrarly large as $n \to \infty$, allowing the test to correctly reject $H_0$. For a fixed level $\alpha$, the test rejects $H_0$ if $\widehat{d}_{\ell_2,\mu,J}^2[X,Y]$ exceeds a predetermined test threshold, which is given by the $(1 - \alpha)$-quantile of its asymptotic null distribution. As it is very computationally costly to obtain quantiles of this distribution, [5] normalize the differences between mean embeddings, and consider instead the test statistic ME[X,Y]:=$\|\sqrt{n}\boldsymbol{\Sigma}_n^{-1/2}\mathbf{S}_n\|_2^2$ where $\mathbf{S}_n := \frac{1}{n}\sum_{i=1}^n \mathbf{z}_i$, $\boldsymbol{\Sigma}_n := \frac{1}{n-1}\sum_{i=1}^n (\mathbf{z}_i - \mathbf{S}_n)(\mathbf{z}_i - \mathbf{S}_n)^T$, and $\mathbf{z}_i := (k(x_i, T_j) - k(y_i, T_j))_{j=1}^J \in \mathbb{R}^J$. Under the null hypothesis $H_0$, asymptotically the ME statistic follows $\chi^2(J)$, a chi-squared distribution with J degrees of freedom. Moreover, for $k$ a translation-invariant kernel, [5] derive another statistical test, called the SCF test (for Smooth Characteristic Function), where its statistic SCF[X,Y] is of the same form as the ME test statistic with a modified $\mathbf{z}_i := [f(x_i)\sin(x_i^T T_j) - f(y_i)\sin(y_i^T T_j), f(x_i)\cos(x_i^T T_j) - f(y_i)\cos(y_i^T T_j)]_{j=1}^J \in \mathbb{R}^{2J}$ where $f$ is the inverse Fourier transform of $k$, and show that under $H_0$, SCF[X,Y] follows asymptotically $\chi^2(2J)$.

## 2  A family of metrics that metrize of the weak convergence

[5] build their ME statistic by estimating the $L^2$ distance between mean embeddings. This metric can be generalized using any $L^p$ distance with $p \geq 1$. These metrics are well suited for the two-sample problem as they metrize the weak convergence (see proof in supp. mat. A.1):

**Theorem 2.1.** *Given $p \geq 1$, $k$ a definite positive, characteristic, continuous, and bounded kernel on $\mathbb{R}^d$, $\mu_P$ and $\mu_Q$ the mean embeddings of the Borel probability measures $P$ and $Q$ respectively, the function defined on $\mathcal{M}_+^1(\mathbb{R}^d) \times \mathcal{M}_+^1(\mathbb{R}^d)$:*

$$d_{L^p,\mu}(P,Q) := \left( \int_{t \in \mathbb{R}^d} \left| \mu_P(t) - \mu_Q(t) \right|^p d\Gamma(t) \right)^{1/p} \tag{7}$$

*is a metric on the space of Borel probability measures, for $\Gamma$ a Borel probability measure absolutely continuous with respect to Lebesgue measure. Moreover a sequence $(\alpha_n)_{n \geq 0}$ of Borel probability measures converges weakly towards $\alpha$ if and only if $d_{L^p,\mu}(\alpha_n, \alpha) \to 0$.*

Therefore, as the MMD, these metrics take into account the geometry of the underlying space and metrize the convergence in law. If we assume in addition that the kernel is analytic, we will show that deriving test statistics from the $L^1$ distance instead of the $L^2$ distance improves the test power for two-sample testing.

## 3  Two-sample testing using the $\ell_1$ norm

### 3.1  A test statistic with simple asymptotic distribution

From now, we assume that $k$ is a positive definite, characteristic, analytic, and bounded kernel.

The statistic presented in eq. 6 is based on the $\ell_2$ norm of a vector that capture differences between distributions in the RKHS at J locations. We will show that using an $\ell_1$ norm instead of an $\ell_2$ norm improves the test power (Proposition 3.1). It captures better the geometry of the problem. Indeed, when $P \neq Q$, the differences between distributions are dense which allow the $\ell_1$ norm to reject better the null hypothesis $H_0$: $P = Q$.

We now build a consistent statistical test based on an empirical estimation of the $L^1$ metric introduced in eq. 7:

$$\widehat{d}_{\ell_1,\mu,J}[X,Y] := \sqrt{n} \sum_{j=1}^J \left| \mu_X(T_j) - \mu_Y(T_j) \right| \tag{8}$$

where $\{T_j\}_{j=1}^J$ are sampled from the distribution $\Gamma$. We show that under $H_0$, $\widehat{d}_{\ell_1,\mu,J}[X,Y]$ converges in distribution to a sum of correlated Nakagami variables[2] and under $H_1$, $\widehat{d}_{\ell_1,\mu,J}[X,Y]$ can be arbitrary large as $n \to \infty$ (see supp. mat. C.1). For a fixed level $\alpha$, the test rejects $H_0$ if $\widehat{d}_{\ell_1,\mu,J}[X,Y]$ exceeds the $(1-\alpha)$-quantile of its asymptotic null distribution. We now compare the power of the statistics based respectively on the $\ell_2$ norm (eq. 6) and the $\ell_1$ norm (eq. 8) at the same level $\alpha > 0$ and we show that the power of the test using the $\ell_1$ norm is better with high probability (see supp. mat. C.2):

**Proposition 3.1.** *Let $\alpha \in ]0,1[$, $\gamma > 0$ and $J \geq 2$. Let $\{T_j\}_{j=1}^J$ sampled i.i.d. from the distribution $\Gamma$ and let $X := \{x_i\}_{i=1}^n$ and $Y := \{y_i\}_{i=1}^n$ i.i.d. samples from $P$ and $Q$ respectively. Let us denote $\delta$ the $(1-\alpha)$-quantile of the asymptotic null distribution of $\widehat{d}_{\ell_1,\mu,J}[X,Y]$ and $\beta$ the $(1-\alpha)$-quantile of the asymptotic null distribution of $\widehat{d}_{\ell_2,\mu,J}^2[X,Y]$. Under the alternative hypothesis, almost surely, there exists $N \geq 1$ such that for all $n \geq N$, with a probability of at least $1 - \gamma$ we have:*

$$\widehat{d}_{\ell_2,\mu,J}^2[X,Y] > \beta \Rightarrow \widehat{d}_{\ell_1,\mu,J}[X,Y] > \delta \qquad (9)$$

Therefore, for a fixed level $\alpha$, under the alternative hypothesis, when the number of samples is large enough, with high probability, the $\ell_1$-based test rejects better the null hypothesis. However, even for fixed $\{T_j\}_{j=1}^J$, computing the quantiles of these distributions requires a computationally-costly bootstrap or permutation procedure. Thus we follow a different approach where we allow the number of samples to differ. Let $X := \{x_i\}_{i=1}^{N_1}$ and $Y := \{y_i\}_{i=1}^{N_2}$ i.i.d according to respectively $P$ and $Q$. We define for any sequence of $\{T_j\}_{j=1}^J$ in $\mathbb{R}^d$:

$$\mathbf{S}_{N_1,N_2} := \Big( \mu_X(T_1) - \mu_Y(T_1), ..., \mu_X(T_J) - \mu_Y(T_J) \Big) \qquad (10)$$

$$\mathbf{Z}_X^i := (k(x_i, T_1), ..., k(x_i, T_J)) \in \mathbb{R}^J \qquad \mathbf{Z}_Y^j := (k(y_j, T_1), ..., k(y_j, T_J)) \in \mathbb{R}^J$$

And by denoting:

$$\boldsymbol{\Sigma}_{N_1} := \frac{1}{N_1 - 1} \sum_{i=1}^{N_1} (\mathbf{Z}_X^i - \overline{\mathbf{Z}}_X)(\mathbf{Z}_X^i - \overline{\mathbf{Z}}_X)^T \qquad \boldsymbol{\Sigma}_{N_2} := \frac{1}{N_2 - 1} \sum_{j=1}^{N_2} (\mathbf{Z}_Y^j - \overline{\mathbf{Z}}_Y)(\mathbf{Z}_Y^j - \overline{\mathbf{Z}}_Y)^T$$

$$\boldsymbol{\Sigma}_{N_1,N_2} := \frac{\boldsymbol{\Sigma}_{N_1}}{\rho} + \frac{\boldsymbol{\Sigma}_{N_2}}{1-\rho}$$

We can define our new statistic as:

$$\text{L1-ME}[X,Y] := \left\| \sqrt{t} \boldsymbol{\Sigma}_{N_1,N_2}^{-\frac{1}{2}} \mathbf{S}_{N_1,N_2} \right\|_1 \qquad (11)$$

We assume that the number of samples of the distributions $P$ and $Q$ are of the same order, i.e: let $t = N_1 + N_2$, we have: $\frac{N_1}{t} \to \rho$ and therefore $\frac{N_2}{t} \to 1 - \rho$ with $\rho \in ]0,1[$. The computation of the statistic requires inverting a $J \times J$ matrix $\boldsymbol{\Sigma}_{N_1,N_2}$, but this is fast and numerically stable: $J$ is typically be small, *eg* less than 10. The next proposition demonstrates the use of this statistic as a consistent two-sample test (see supp. mat. C.3 for the proof).

**Proposition 3.2.** *Let $\{T_j\}_{j=1}^J$ sampled i.i.d. from the distribution $\Gamma$ and $X := \{x_i\}_{i=1}^{N_1}$ and $Y := \{y_i\}_{i=1}^{N_2}$ be i.i.d. samples from $P$ and $Q$ respectively. Under $H_0$, the statistic L1-ME$[X,Y]$ is almost surely asymptotically distributed as Naka$(\frac{1}{2}, 1, J)$, a sum of $J$ random variables i.i.d which follow a Nakagami distribution of parameter $m = \frac{1}{2}$ and $\omega = 1$. Finally under $H_1$, almost surely the statistic can be arbitrarily large as $t \to \infty$, enabling the test to correctly reject $H_0$.*

**Statistical test of level $\alpha$:** Compute $\| \sqrt{t} \boldsymbol{\Sigma}_{N_1,N_2}^{-\frac{1}{2}} \mathbf{S}_{N_1,N_2} \|_1$, choose the threshold $\delta$ corresponding to the $(1-\alpha)$-quantile of Naka$(\frac{1}{2}, 1, J)$, and reject the null hypothesis whenever $\| \sqrt{t} \boldsymbol{\Sigma}_{N_1,N_2}^{-\frac{1}{2}} \mathbf{S}_{N_1,N_2} \|_1$ is larger than $\delta$.

$f(x, m, \omega) = \frac{2m^m}{\Gamma(m)\omega^m} x^{2m-1} \exp(\frac{-m}{\omega} x^2)$ where $\Gamma$ is the Gamma function.

## 3.2 Optimizing test locations to improve power

As in [17], we can optimize the test locations $\mathcal{V}$ and kernel parameters (jointly referred to as $\theta$) by maximizing a lower bound on the test power which offers a simple objective function for fast parameter tuning. We make the same regularization as in [17] of the test statistic for stability of the matrix inverse, by adding a regularization parameter $\gamma_{N_1,N_2} > 0$ which goes to 0 as $t$ goes to infinity, giving L1-ME$[X, Y] := \|\sqrt{t}(\boldsymbol{\Sigma}_{N_1,N_2} + \gamma_{N_1,N_2}\mathbf{I})^{-\frac{1}{2}}\mathbf{S}_{N_1,N_2}\|_1$ (see proof in supp. mat. D.1).

**Proposition 3.3.** *Let $\mathcal{K}$ be a uniformly bounded family of $k : \mathbb{R}^d \times \mathbb{R}^d \to \mathbb{R}$ measurable kernels (i.e., $\exists K < \infty$ such that $\sup_{k \in \mathcal{K}} \sup_{(x,y) \in (\mathbb{R}^d)^2} |k(x,y)| \le K$). Let $\mathcal{V}$ be a collection in which each element is a set of $J$ test locations. Assume that $c := \sup_{V \in \mathcal{V}, k \in \mathcal{K}} \|\boldsymbol{\Sigma}^{-1/2}\| < \infty$. Then the test power $\mathbb{P}\left(\widehat{\lambda}_t \ge \delta\right)$ of the L1-ME test satisfies $\mathbb{P}\left(\widehat{\lambda}_t \ge \delta\right) \ge L(\lambda_t)$ where:*

$$L(\lambda_t) = 1 - 2\sum_{k=1}^{J} \exp\left(-\left(\frac{\lambda_t - \delta}{J^2 + J}\right)^2 \frac{\gamma_{N_1,N_2} N_1 N_2}{(N_1 + N_2)^2}\right)$$

$$- 2\sum_{k,q=1}^{J} \exp\left(-2\frac{\left(\frac{\gamma_{N_1,N_2}}{K_3 J^2}\frac{\lambda_t - \delta}{(J^2 + J)\sqrt{t}} - \frac{J^3 K_2}{\sqrt{\gamma_{N_1,N_2}}} - J^4 K_1\right)^2}{K_\lambda^2 (N_1 + N_2)\max\left(\frac{8}{\rho N_1}, \frac{8}{(1-\rho)N_2}\right)^2}\right)$$

*and $K_1$, $K_2$, $K_3$ and $K_\lambda$, are positive constants depending on only $K$, $J$ and $c$. The parameter $\lambda_t := \|\sqrt{t}\boldsymbol{\Sigma}^{-\frac{1}{2}}\mathbf{S}\|_1$ is the population counterpart of $\widehat{\lambda}_t := \|\sqrt{t}(\Sigma_{N_1,N_2} + \gamma_{N_1,N_2}\mathbf{I})^{-\frac{1}{2}}\mathbf{S}_{N_1,N_2}\|_1$ where $\mathbf{S} = \mathbb{E}_{x,y}(S_{N_1,N_2})$ and $\boldsymbol{\Sigma} = \mathbb{E}_{x,y}(\boldsymbol{\Sigma}_{N_1,N_2})$. Moreover for large $t$, $L(\lambda_t)$ is increasing in $\lambda_t$.*

Proposition 3.3 suggests that it is sufficient to maximize $\lambda_t$ to maximize a lower bound on the L1-ME test power. The statistic $\lambda_t$ for this test depends on a set of test locations $\mathcal{V}$ and a kernel parameter $\sigma$. We set $\theta^* := \{\mathcal{V}, \sigma\} = \arg\max_\theta \lambda_t = \arg\max_\theta \|\sqrt{t}\,\boldsymbol{\Sigma}^{-\frac{1}{2}}\mathbf{S}\|_1$. As proposed in [14], we can maximize a proxy test power to optimize $\theta$: it does not affect $H_0$ and $H_1$ as long as the data used for parameter tuning and for testing are disjoint.

## 3.3 Using smooth characteristic functions (SCF)

As the ME statistic, the SCF statistic estimates the $L^2$ distance between well chosen distribution representatives. Here, the representatives of the distributions are the convolution of their characteristic functions and the kernel $k$, assumed translation-invariant. [5] use them to detect differences between distributions in the frequency domain. We show that the $L^1$ version (denoted $d_{L^1,\Phi}$) is a metric on the space of Borel probability measures with integrable characteristic functions such that if $\alpha_n$ converge weakly towards $\alpha$, then $d_{L^1,\Phi}(\alpha_n, \alpha) \to 0$ (see supp. mat. A.2). Let us introduce the test statistics in the frequency domain respectively based on the $\ell_2$ norm and on the $\ell_1$ norm which lead to consistent tests:

$$\widehat{d}_{\ell_2,\Phi,J}^2[X,Y] := \|\sqrt{n}\mathbf{S}_n\|_2^2 \quad \text{and} \quad \widehat{d}_{\ell_1,\Phi,J}[X,Y] := \|\sqrt{n}\mathbf{S}_n\|_1 \tag{12}$$

where $\mathbf{S}_n := \frac{1}{n}\sum_{i=1}^{n}\mathbf{z}_i$, $\mathbf{z}_i := [f(x_i)\sin(x_i^T T_j) - f(y_i)\sin(y_i^T T_j), f(x_i)\cos(x_i^T T_j) - f(y_i)\cos(y_i^T T_j)]_{j=1}^J \in \mathbb{R}^{2J}$ and $f$ is the inverse Fourier transform of $k$. We show that, at the same level $\alpha$, using the $\ell_1$ norm in the frequency domain provides a better power with high probability (see supp. mat. E.1):

**Proposition 3.4.** *Let $\alpha \in ]0,1[$, $\gamma > 0$ and $J \ge 2$. Let $\{T_j\}_{j=1}^J$ sampled i.i.d. from the distribution $\Gamma$ and let $X := \{x_i\}_{i=1}^n$ and $Y := \{y_i\}_{i=1}^n$ i.i.d. samples from $P$ and $Q$ respectively. Let us denote $\delta$ the $(1-\alpha)$-quantile of the asymptotic null distribution of $\widehat{d}_{\ell_1,\Phi,J}[X,Y]$ and $\beta$ the $(1-\alpha)$-quantile of the asymptotic null distribution of $\widehat{d}_{\ell_2,\Phi,J}^2[X,Y]$. Under the alternative hypothesis, almost surely, there exists $N \ge 1$ such that for all $n \ge N$, with a probability of at least $1 - \gamma$ we have:*

$$\widehat{d}_{\ell_2,\Phi,J}^2[X,Y] > \beta \Rightarrow \widehat{d}_{\ell_1,\Phi,J}[X,Y] > \delta \tag{13}$$

We now adapt the construction of the L1-ME test to the frequency domain to avoid computational issues of the quantiles of the asymptotic null distribution:

$$\text{L1-SCF}[X, Y] := \|\sqrt{t}\, \boldsymbol{\Sigma}_{N_1,N_2}^{-\frac{1}{2}} \mathbf{S}_{N_1,N_2}\|_1 \tag{14}$$

with $\boldsymbol{\Sigma}_{N_1,N_2}$, and $\mathbf{S}_{N_1,N_2}$ defined as in the L1-ME statistic with new expression for $\mathbf{Z}_X^i$ (and $\mathbf{Z}_Y^j$):

$$\mathbf{Z}_X^i = \left( cos\left(T_1^T x_i\right) f(x_i), ..., sin\left(T_J^T x_i\right) f(x_i)\right) \in \mathbb{R}^{2J}$$

From this statistic, we build a consistent test. Indeed, an analogue proof of the Proposition 3.2 gives that under $H_0$, L1-SCF$[X, Y]$ is a.s. asymptotically distributed as Naka($\frac{1}{2}$, 1, 2$J$), and under $H_1$, the test statistic can be arbitrarily large as t goes to infinity. Finally an analogue proof of Proposition 3.3 shows that we can optimize the test locations and the kernel parameter to improve the power as well.

## 4 Experimental study

We now run empirical comparisons of our $\ell_1$-based tests to their $\ell_2$ counterparts, state-of-the-art Kernel-based two-sample tests. We study both toy and real problems. We use the isotropic Gaussian kernel class $\mathcal{K}_g$. We call **L1-opt-ME** and **L1-opt-SCF** the tests based respectively on mean embeddings and smooth characteristic functions proposed in this paper when optimizing test locations and the Gaussian width $\sigma$ on a separate training set of the same size as the test set. We denote also **L1-grid-ME** and **L1-grid-SCF** where only the Gaussian width is optimized by a grid search, and locations are randomly drawn from a multivariate normal distribution. We write **ME-full** and **SCF-full** for the tests of [17], also fully optimized according to their criteria. **MMD-quad** (quadratic-time) and **MMD-lin** (linear-time) refer to the MMD-based tests of [11], where, to ensure a fair comparison, the kernel width is also set to maximize the test power following [14]. For **MMD-quad**, as its null distribution is an infinite sum of weighted chi-squared variables (no closed-form quantiles), we approximate the null distribution with 200 random permutations in each trial.

In all the following experiments, we repeat each problem 500 times. For synthetic problems, we generate new samples from the specified $P$, $Q$ distributions in each trial. For the first real problem (Higgs dataset), as the dataset is big enough we use new samples from the two distributions for each trial. For the second and third real problem (Fast food and text datasets), samples are split randomly into train and test sets in each trial. In all the simulations we report an empirical estimate of the Type-I error when $H_0$ hold and of the Type-II error when $H_1$ hold. We set $\alpha = 0.01$. The code is available at https://github.com/meyerscetbon/l1_two_sample_test.

**How to realize $\ell_1$-based tests ?** The asymptotic distributions of the statistics is a sum of i.i.d. Nakagami distribution. [8] give a closed form for the probability density function. As the formula is not simple, we can also derive an estimate of the CDF (see supp. mat. F.1).

**Optimization** For a fair comparison between our tests and those of [17], we use the same initialization of the test locations[3]. For the ME-based tests, we initialize the test locations with realizations from two multivariate normal distributions fitted to samples from $P$ and $Q$ and for the for initialization of the SCF-based tests, we use the standard normal distribution. The regularization parameter is set to $\gamma_{N_1,N_2} = 10^{-5}$. The computation costs for our proposed tests are the same as that of [17]: with $t$ samples, optimization is $\mathcal{O}(J^3 + dJt)$ per gradient ascent iteration and testing $\mathcal{O}(J^3 + Jt + dJt)$ (see supp. mat. Table 3).

The experiments on synthetic problems mirror those of [17] to make a fair comparison between the prior art and the proposed methods.

**Test power vs. sample size** We consider four synthetic problems: Same Gaussian (SG, dim= 50), Gaussian mean difference (GMD, dim= 100), Gaussian variance difference (GVD, dim= 30), and Blobs. Table 1 summarizes the specifications of $P$ and $Q$. In the Blobs problem, $P$ and $Q$ are a mixture of Gaussian distributions on a $4 \times 4$ grid in $\mathbb{R}^2$. This problem is challenging as the difference of $P$ and $Q$ is encoded at a much smaller length scale compared to the global structure as explained in [14]. We set $J = 5$ in this experiment.

| Data | $P$ | $Q$ |
|------|-----|-----|
| SG | $\mathcal{N}(0, I_d)$ | $\mathcal{N}(0, I_d)$ |
| GMD | $\mathcal{N}(0, I_d)$ | $\mathcal{N}((1, 0, .., 0)^T, I_d)$ |
| GVD | $\mathcal{N}(0, I_d)$ | $\mathcal{N}(0, diag(2, 1, .., 1))$ |
| Blobs | Mixture of 16 Gaussians in $\mathbb{R}^2$ as [17] | |

Table 1: Synthetic problems. $H_0$ holds only in SG.

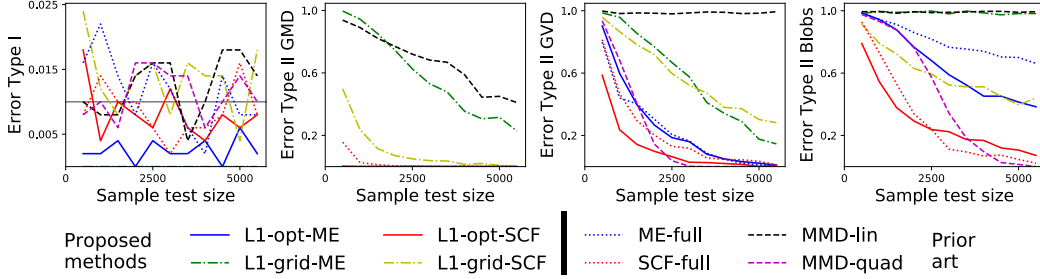

Figure 1: Plots of type-I/type-II errors against the test sample size $n^{te}$ in the four synthetic problems.

Figure 1 shows type-I error (for SG problem), and test power (for GMD, GVD and Blobs problem) as a function of test sample size. In the SG problem, the type-I error roughly stays at the specified level $\alpha$ for all tests except the L1-ME tests, which reject the null at a rate below the specified level $\alpha$. Therefore, here these tests are more conservative.

GMD with 100 dimensions is an easy problem for **L1-opt-ME**, **L1-opt-SCF**, **ME-full MMD-quad**, while the **SCF-full** test requires many samples to achieve optimal test power. In the GMD, GVD and Blobs cases, **L1-opt-ME** and **L1-opt-SCF** achieve substantially higher test power than **L1-grid-ME** and **L1-grid-SCF**, respectively: optimizing the test locations brings a clear benefit. Remarkably **L1-opt-SCF** consistently outperforms the quadratic-time **MMD-quad** up to 2 500 samples in the GVD case. SCF variants perform significantly better than ME variants on the Blobs problem, as the difference in $P$ and $Q$ is localized in the frequency domain. For the same reason, **L1-opt-SCF** does much better than the quadratic-time MMD up to 3 000 samples, as the latter represents a weighted distance between characteristic functions integrated across the frequency domain as explained in [29].

We also perform a more difficult GMD problem to distinguish the power of the proposed tests with the **ME-full** as all reach maximal power. **L1-opt-ME** then performs better than **ME-full**, its $\ell_2$ counterpart, as it needs less data to achieve good control (see mat. supp. F.3).

**Test power vs. dimension d** On fig 2, we study how the dimension of the problem affects type-I error and test power of our tests. We consider the same synthetic problems: SG, GMD and GVD, we fix the test sample size to 10000, set $J = 5$, and vary the dimension. Given that these experiments explore large dimensions and a large number of samples, computing the **MMD-quad** was too expensive.

In the SG problem, we observe the **L1-ME** tests are more conservative as dimension increases, and the others tests can maintain type-I error at roughly the specified significance level $\alpha = 0.01$. In the GMD problem, we note that the tests proposed achieve the maximum test power without making error of type-II whatever the dimension is, while the **SCF-full** loses power as dimension increases. However, this is true only with optimization of the test locations as it is shown by the test power of **L1-grid-ME** and **L1-grid-SCF** which drops as dimension increases. Moreover the performance of **MMD-lin** degrades quickly with increasing dimension, as expected from [25]. Finally in the GVD problem, all tests failed to keep a good test power as the dimension increases, except the **L1-opt-SCF**, which has a very low type-II for all dimensions. These results echo those obtained by [34]. Indeed [34] study a class of two sample test statistics based on inter-point distances and they show benefits of using the $\ell_1$ norm over the Euclidean distance and the Maximum Mean Discrepancy (MMD) when the dimensionality goes to infinity. For this class of test statistics, they characterize asymptotic power

Figure 2: Plots of type-I/type-II error against the dimension in three synthetic problems: SG (Same Gaussian), GMD (Gaussian Mean Difference), and GVD (Gaussian Variance Difference).

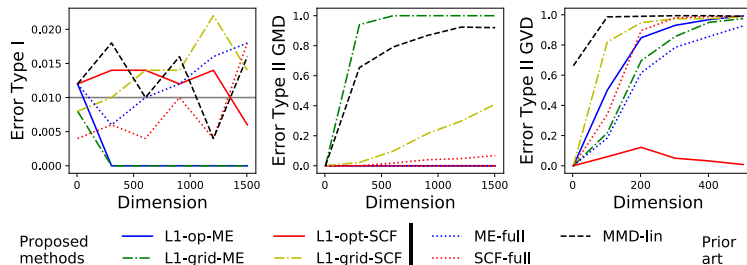

loss w.r.t the dimension and show that the $\ell_1$ norm is beneficial compared to the $\ell_2$ norm provided that the summation of discrepancies between marginal univariate distributions is large enough.

**Informative features** Figure 3 we replicate the experiment of [17], showing that the selected locations capture multiple modes in the $\ell_1$ case, as in the $\ell_2$ case. (details in supp. mat. F.4). The figure shows that the objective function $\widehat{\lambda}_t^{tr}(T_1, T_2)$ used to position the second test location $T_2$ has a maximum far from the chosen position for the first test location $T_1$.

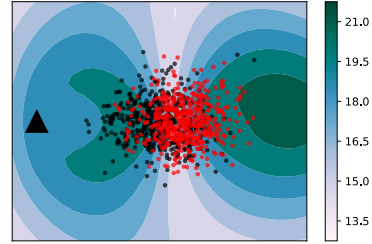

**Real Data 1, Higgs**: The first real problem is the Higgs dataset [21] described in [2]: distinguishing signatures of Higgs bosons from the background. We use a two-sample test on 4 derived features as in [5]. We compare for various sample sizes the performance of the proposed tests with those of [17]. We do not study the **MMD-quad** test as its computation is too expensive with 10 000 samples. To make the problem harder, we only consider $J = 3$ locations. Fig. 4 shows a clear benefit of the optimized $\ell_1$-based tests, in particular for SCF (**L1-opt-SCF**) compared to its $\ell_2$ counterpart (**SCF-full**). Optimizing the location is important, as **L1-opt-SCF** and **L1-opt-ME** perform much better than their grid versions (which are comparable to the tests of [5]).

Figure 3: **Illustrating interpretable features**, replicating in the $\ell_1$ case the figure of [17]. A contour plot of $\widehat{\lambda}_t^{tr}(T_1, T_2)$ as a function of $T_2$, when $J = 2$, and $T_1$ is fixed. The red and black dots represent the samples from the $P$ and $Q$ distributions, and the big black triangle the position of $T_1$ –complete figure in supp. mat. F.4.

**Real Data 2, Fastfood:** We use a Kaggle dataset listing locations of over 10,000 fast food restaurants across America[4]. We consider the 6 most frequent brands in mainland USA: Mc Donald's, Burger King, Taco Bell, Wendy's, Arby's and KFC. We benchmark the various two-sample tests to test whether the spatial distribution (in $\mathbb{R}^2$) of restaurants differs across brand. This is a non trivial question, as it depends on marketing strategy of the brand. We compare the distribution of Mc Donald's restaurants with others. We also compare the distribution of Mc Donald's restaurants with itself to evaluate the level of the tests (see supp. mat. Table 5). The number of samples differ across the distributions; hence to perform the tests from [17], we randomly subsample the largest distribution. We use $J = 3$ as the number of locations.

Figure 4: **Higgs dataset**: Plots of type-II errors against the test sample size $n^{te}$.

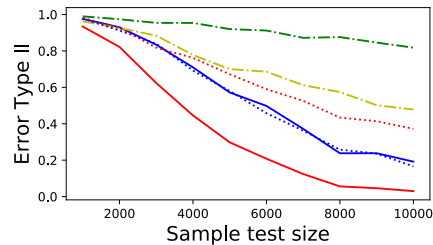

| Proposed methods | Prior art |
|---|---|
| — L1-opt-ME — L1-opt-SCF ⋯⋯ ME-full | ⋯⋯ SCF-full |
| -·- L1-grid-ME -·- L1-grid-SCF | |

| Problem | L1-opt-ME | L1-grid-ME | L1-opt-SCF | L1-grid-SCF | ME-full | SCF-full | MMD-quad |
|---|---|---|---|---|---|---|---|
| McDo vs Burger King (1141) | 0.112 | 0.426 | 0.428 | 0.960 | 0.170 | 0.094 | 0.184 |
| McDo vs Taco Bell (877) | 0.554 | 0.624 | 0.710 | 0.834 | 0.684 | 0.638 | 0.666 |
| McDo vs Wendy's (733) | 0.156 | 0.246 | 0.752 | 0.942 | 0.416 | 0.624 | 0.208 |
| McDo vs Arby's (517) | 0.000 | 0.004 | 0.006 | 0.468 | 0.004 | 0.012 | 0.004 |
| McDo vs KFC (429) | 0.912 | 0.990 | 1.00 | 0.998 | 0.996 | 0.856 | 0.980 |

Table 2: **Fast food dataset:** Type-II errors for distinguishing the distribution of fast food restaurants. $\alpha = 0.01$. $J = 3$. The number in brackets denotes the sample size of the distribution on the right. We consider MMD-quad as the gold standard.

Table 2 summarizes type-II errors of the tests. Note that it is not clear that distributions must differ, as two brands sometimes compete directly, and target similar locations. We consider the **MMD-quad** as the gold standard to decide whether distributions differ or not. The three cases for which there seems to be a difference are Mc Donald's vs Burger King, Mc Donald's vs Wendy's, and Mc Donalds vs Arby's. Overall, we find that the optimized **L1-opt-ME** agrees best with this gold standard. The Mc Donald's vs Arby's problem seems to be an easy problem, as all tests reach a maximal power, except for the **L1-grid-SCF** test which shows the gain of power brought by the optimization. In the Mc Donald's vs Wendy's problem the **L1-opt-ME** test outperforms the $\ell_2$ tests and even the quadratic-time MMD. Finally, all the tests fail to discriminate Mc Donald's vs KFC. The data provide no evidence that these brands pursue different strategies to chose locations.

In the Mc Donald's vs Burger King and Mc Donald's vs Wendy's problems, the optimized version of the test proposed based on mean embedding outperform the grid version. This success implies that the locations learned are each informative, and we plot them (see supp. mat. Figure 8), to investigate the interpretability of the **L1-opt-ME** test. The figure shows that the procedure narrows on specific regions of the USA to find differences between distributions of restaurants.

**Real Data 3, text:** For a high-dimension problem, we consider the problem of distinguishing the newsgroups text dataset [18] (details in supp. Mat. F.5). Compared to their $\ell_2$ counterpart, $\ell_1$-optimized tests bring clear benefits and separate all topics of articles based on their word distribution.

**Discussion:** Our theoretical results suggest it is always beneficial for statistical power to build tests on $\ell_1$ norms rather than $\ell_2$ norm of differences between kernel distribution representatives (Propositions 3.1, 3.4). In practice, however, optimizing test locations with $\ell_1$-norm tests leads to non-smooth objective functions that are harder to optimize. Our experiments confirm the theoretical benefit of the $\ell_1$-based framework. The benefit is particularly pronounced for a large number $J$ of test locations –as the difference between $\ell_1$ and $\ell_2$ norms increases with dimension (see in supp. mat. Lemmas 8, 12)– as well as for large dimension of the native space (Figure 2). The benefit of $\ell_1$ distances for two-sample testing in high dimension has also been reported by [34], though their framework does not link to kernel embeddings or to the convergence of probability measures. Further work should consider extending these results to goodness-of-fit testing, where the $L^1$ geometry was shown empirically to provide excellent performance [16].

## 5 Conclusion

In this paper, we show that statistics derived from the $L^p$ distances between well-chosen distribution representatives are well suited for the two-sample problem as these distances metrize the weak convergence (Theorem 2.1). We then compare the power of tests introduced in [5] and their $\ell_1$ counterparts and we show that $\ell_1$-based statistics have better power with high probability (Propositions 3.1, 3.4). As with state-of-the-art Euclidean approaches, the framework leads to tractable computations and learns interpretable locations of where the distributions differ. Empirically, on all 4 synthetic and 3 real problems investigated, the $\ell_1$ geometry gives clear benefits compared to the Euclidean geometry. The $L^1$ distance is known to be well suited for densities, to control differences or estimation [7]. It is also beneficial for kernel embeddings of distributions.

**Acknowledgments** This work was funded by the DirtyDATA ANR grant (ANR-17-CE23-0018). We also would like to thank Zoltán Szabó from École Polytechnique for crucial suggestions, and acknowledge hardware donations from NVIDIA Corporation.

## Footnotes

[1] A random metric is a random process which satisfies all the conditions for a metric 'almost surely' [5].

[5] show that for $\{T_j\}_{j=1}^J$ sampled from the distribution $\Gamma$, under the null hypothesis $H_0 : P = Q$, as $n \to \infty$, the following test statistic:

[2]the pdf of the Nakagami distribution of parameters $m \geq \frac{1}{2}$ and $\omega > 0$ is $\forall x \geq 0$,

[4]www.kaggle.com/datafiniti/fast-food-restaurants

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
