[Supplementary Material]

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

# Supplementary materials

## Table of Contents

## A  A family of metrics that metrize of the weak convergence

### A.1  Distances between Mean Embeddings

**Theorem 1.** *Given $p \geq 1$, $k$ a definite positive, characteristic, continuous, and bounded kernel on $\mathbb{R}^d$, $\mu_P$ and $\mu_Q$ the mean embeddings of the Borel probability measures $P$ and $Q$ respectively, the function defined on $\mathcal{M}^1_+(\mathbb{R}^d) \times \mathcal{M}^1_+(\mathbb{R}^d)$:*

$$d_{L^p,\mu}(P,Q) := \left( \int_{t \in \mathbb{R}^d} \left| \mu_P(t) - \mu_Q(t) \right|^p d\Gamma(t) \right)^{1/p} \tag{15}$$

*is a metric on the space of Borel probability measures, for $\Gamma$ a Borel probability measure absolutely continuous with respect to Lebesgue measure. Moreover a sequence $(\alpha_n)_{n \geq 0}$ of Borel probability measures converges weakly towards $\alpha$ if and only if $d_{L^p,\mu}(\alpha_n, \alpha) \to 0$.*

**Proof.** *First, let us prove that for any $p \geq 1$ $d_{L^p,\mu}$ is metric of on the space of Borel probability measures. Let $p \geq 1$, we have:*

$$|\mu_P(t) - \mu_Q(t)|^p = |\langle \mu_P - \mu_Q, k_t \rangle|^p$$

*Therefore:*

$$|\mu_P(t) - \mu_Q(t)|^p \leq ||\mu_P - \mu_Q||_H^p (k(t,t))^{p/2}$$

*But as $k$ is bounded, and $\Gamma$ is finite, $d_{L^p,\mu}$ is well defined on $\mathcal{M}^1_+(\mathcal{X})^2$. Let us prove now that if $P \neq Q$ then $d_{L^p,\mu}(P,Q) > 0$.*

**Definition 1.** *[10] A kernel is characteristic if the mapping $P \in \mathcal{M}_+^1(\mathcal{X}) \to \mu_P \in H_k$ is injective, where $H_k$ is the RKHS associated with $k$.*

**Lemma 1.** *[31] If k is a continuous kernel on a metric space then every feature maps associated with the kernel are continuous.*

*Let $P$ and $Q$ two Borel distributions such that $P \neq Q$. Since the mapping $p \to \mu_P$ is injective, there must exists at least one point $o$ where $\mu_P - \mu_Q$ is non-zero. By continuity of $\mu_P - \mu_Q$, there exists a ball around $o$ in which $\mu_P - \mu_Q$ is non-zero. Then $d_{L^p,\mu}(P,Q) > 0$. Finally all the other proprieties of a metric are clearly verified by this function.*

*Let us now show that $d_{L^1,\mu}$ metrize the weak convergence. For that purpose, we first show that this metric has an IPM formulation:*

**Lemma 2.** *l We denote by $\mathcal{T}_k$ the integral operator on $L_2^{d\Gamma}(\mathbb{R}^d)$ associated to the positive definite, characteristic, continuous, and bounded kernel $k$ defined as:*

$$\mathcal{T}_k \quad : \quad \begin{array}{ccc} L_2^{d\Gamma}(\mathbb{R}^d) & \to & L_2^{d\Gamma}(\mathbb{R}^d) \\ f & \to & \int_{\mathbb{R}^d} k(x,.)f(x)d\Gamma(x) \end{array}$$

*By denoting $B_\infty^{d\Gamma}$ the unit ball of $L_\infty^{d\Gamma}(\mathbb{R}^d)$, we have that:*

$$d_{L^1,\mu}(P,Q) = \sup_{f \in \mathcal{T}_k(B_\infty^{d\Gamma})} \left( \mathbb{E}_P[f(X)] - \mathbb{E}_Q[f(Y)] \right)$$

**Proof.** *We have:*

$$d_{L^1,\mu}(P,Q) = \int_{x \in \mathbb{R}^d} |\mu_P(x) - \mu_Q(x)| d\Gamma(x)$$

$$= \int_{x \in \mathbb{R}^d} |\langle \mu_P - \mu_Q, k_x \rangle_H| d\Gamma(x)$$

$$= \int_{x \in \{v:\mu_P(v) \geq \mu_Q(v)\}} \langle \mu_P - \mu_Q, k_x \rangle_H d\Gamma(x) - \int_{x \in \{v:\mu_P(v) < \mu_Q(v)\}} \langle \mu_P - \mu_Q, k_x \rangle_H d\Gamma(x)$$

$$= \langle \mu_P - \mu_Q, \int_{x \in \{v:\mu_P(v) \geq \mu_Q(v)\}} k_x d\Gamma(x) - \int_{x \in \{v:\mu_P(v) < \mu_Q(v)\}} k_x d\Gamma(x) \rangle_H$$

*Then:*

$$d_{L^1,\mu}(P,Q) = \langle \mu_P - \mu_Q, f \rangle_H$$

*with*

$$f = \int_{t \in \mathbb{R}^d} g_t d\Gamma(t) \quad where \quad g_t = \left\{ \begin{array}{l} k_t \ if \ t \in \{x : \mu_P(x) \geq \mu_Q(x)\} \\ -k_t \ otherwise. \end{array} \right. \tag{16}$$

*Therefore, $f \in T_k(B_\infty^{d\Gamma}) \subset H_k$ and we have:*

$$d_{L^1,\mu}(P,Q) = \mathbb{E}_P(f(X)) - \mathbb{E}_Q(f(Y))$$

*Now, let $f$ be an element of $\mathcal{T}_k(B_\infty^{d\Gamma}) \subset H_k$. Therefore there exists $g \in B_\infty^{d\Gamma}$ such that $f = \mathcal{T}_k(g)$ and we have then:*

$$\mathbb{E}_P(f(X)) - \mathbb{E}_Q(f(Y)) = \langle \mu_P - \mu_Q, f \rangle$$

$$= \langle \mu_P - \mu_Q, \int_{t \in \mathbb{R}^d} g(t)k_t d\Gamma(t) \rangle$$

$$= \int_{t \in \mathbb{R}^d} g(t)\langle \mu_P - \mu_Q, k_t \rangle d\Gamma(t)$$

$$= \int_{t \in \mathbb{R}^d} g(t)(\mu_P(t) - \mu_Q(t)) d\Gamma(t)$$

$$\leq \int_{t \in \mathbb{R}^d} |\mu_P(t) - \mu_Q(t)| d\Gamma(t)$$

*Therefore we have:*

$$d_{L^1,\mu}(P,Q) = \sup_{f \in \mathcal{T}_k(B_\infty^{d\Gamma})} \left( \mathbb{E}_P[f(X)] - \mathbb{E}_Q[f(Y)] \right)$$

*From this IPM formulation we now show that $d_{L^1,\mu}$ metrize the weak convergence. First, as the kernel $k$ is assumed to be continuous, then $\mathcal{T}_k(B_\infty^{d\Gamma}) \subset H_k \subset C^0(\mathbb{R}^d)$, the set of continuous functions. Therefore, thanks to the IPM formulation of the metric, the weak convergence of a sequence of distributions $(\alpha_n)_{n\geq 0}$ towards a distribution $\alpha$ implies the convergence according to the $d_{L^1,\mu}$-distance. Conversely let $\alpha \in \mathcal{M}_+^1(\mathcal{X})$ and let us assume that $(\alpha_n)_{n\geq 0}$ is a sequence of Borel probability measures such that $d_{L^1,\mu}(\alpha_n, \alpha) \to 0$. Since $\int\limits_{x\in\mathbb{R}^d} k(x,x)d\Gamma(x)$ is finite, $T_k$ is self-adjoint, positive semi-definite and trace-class [27]. It has at most countably many positive eigenvalues $(\lambda_m)_{m\geq 0}$ and corresponding orthonormal eigenfunctions $(e_m)_{m\geq 0}$. Then the Mercer theorem [6] gives that $(\lambda_m^{1/2} e_m)_{m\geq 0}$ is an orthonormal basis of $H_k$. Let us denote $C = \sup\limits_{x\in\mathbb{R}^d} \sqrt{k(x,x)}$ And $V_m = \frac{\lambda_m^{1/2} e_m}{C}$. Therefore we have:*

$$\|V_m\|_{\infty,d\Gamma} \leq \frac{\|\lambda_m^{1/2} e_m\|_{H_k}}{C} C \leq 1$$

*Therefore, thanks to Lemma 2, for all $m \geq 0$, we have:*

$$\langle \mu_{\alpha_n} - \mu_\alpha, T_k(V_m)\rangle_{H_k} \to 0$$

*Now, we want to show that for every $f \in H_k$, $\langle \mu_{\alpha_n} - \mu_\alpha, f\rangle_{H_k} \to 0$. Let us consider $f \in H_k$. As $(\lambda_m^{1/2} e_m)_{m\geq 0}$ is an orthonormal basis of $H_k$, we have:*

$$f = \sum_{m\geq 0} \langle f, \lambda_m^{1/2} e_m\rangle_{H_k} \lambda_m^{1/2} e_m$$

*Therefore if we define for every $m \geq 0$:*

$$f_m := \sum_{i=0}^{m} \langle f, \lambda_i^{1/2} e_i\rangle_{H_k} \lambda_i^{1/2} e_i$$

*We have that:*

$$\|f_m - f\|_{H_k} \to 0$$

*Therefore let $\epsilon > 0$, and $K$ such that:*

$$\|f_K - f\|_{H_k} \leq \epsilon$$

*First we remarks that:*

$$\langle \mu_{\alpha_n} - \mu_\alpha, f_K\rangle = \sum_{i=0}^{K} \langle f, \lambda_i^{1/2} e_i\rangle \langle \mu_{\alpha_n} - \mu_\alpha, \lambda_i^{1/2} e_i\rangle$$

$$= \sum_{i=0}^{K} \langle f, \lambda_i^{1/2} e_i\rangle C \langle \mu_{\alpha_n} - \mu_\alpha, V_i\rangle$$

$$= \sum_{i=0}^{K} \langle f, \lambda_i^{1/2} e_i\rangle \frac{C}{\lambda_i} \langle \mu_{\alpha_n} - \mu_\alpha, T_k(V_i)\rangle$$

*Indeed the last equality hold as all the eigenvalues are positives. Finally we have that:*

$$\langle \mu_{\alpha_n} - \mu_\alpha, f_K\rangle_{H_k} \to 0 \text{ as } n \text{ goes to infinity.}$$

*Let $N$, such that for $n \geq N$:*

$$\langle \mu_{\alpha_n} - \mu_\alpha, f_K\rangle_{H_k} \leq \epsilon$$

*Therefore we have for all $n \geq N$:*

$$\langle \mu_{\alpha_n} - \mu_\alpha, f\rangle = \langle \mu_{\alpha_n} - \mu_\alpha, f_K\rangle + \langle \mu_{\alpha_n} - \mu_\alpha, f - f_K\rangle$$
$$\leq \epsilon + \|\mu_{\alpha_n} - \mu_\alpha\|_{H_k} \|f - f_K\|_{H_k}$$
$$\leq \epsilon + \|\mu_{\alpha_n} - \mu_\alpha\|_{H_k} \epsilon$$

*Finally as $k$ is bounded, we have that:*

$$\|\mu_{\alpha_n} - \mu_\alpha\|_{H_k} \leq 2 \sup_{x,t} \sqrt{k(x,t)}$$

*Finally we have that for every $f \in H_k$:*

$$\langle \mu_{\alpha_n} - \mu_\alpha, f \rangle \to 0$$

*Therefore for any $f \in B_{H_k}$, the unit ball of the RKHS, we have:*

$$\langle \mu_{\alpha_n} - \mu_\alpha, f \rangle \to 0$$

*And then:*

$$MMD[\alpha_n, \alpha] \to 0$$

*Moreover we have the following theorem:*

**Theorem 2.** *([28]) A bounded kernel over a locally compact Hausdorff space $\mathcal{X}$ metrizes the weak convergence of probability measures iff it is continuous and characteristic.*

*Therefore $\alpha_n$ converge weakly towards $\alpha$ and $d_{L^1,\mu}$ metrize the weak convergence. Moreover thanks to Hölder's inequality we have that for any $p \geq 1$:*

$$d_{L^1,\mu}(P,Q) \leq d_{L^p,\mu}(P,Q) \tag{17}$$

*Moreover as the kernel $k$ is bounded we have also:*

$$d_{L^p,\mu}(P,Q)^p \leq \|\mu_P - \mu_Q\|_\infty^{p-1} d_{L^1,\mu}(P,Q) \tag{18}$$

$$\leq (2C^2)^{p-1} d_{L^1,\mu}(P,Q) \tag{19}$$

*Therefore for any $p \geq 1$ $d_{L^p,\mu}$ metrizes the weak convergence.*

## A.2 Distances between Smooth Characteristic Functions

**Definition 2.** *[5] Let $k : \mathbb{R}^d \to \mathbb{R}$ be a translation-invariant kernel i.e., $k(x-y)$ defines a positive definite kernel for $x$ and $y$, $P$ a Borel probability measure and $\psi_P(t) := \mathbb{E}_x \left( \exp(ix^T t) \right)$ be the characteristic function of $P$. A smooth characteristic function $\Phi_P$ is defined as:*

$$\Phi_P(v) := \int_{\mathbb{R}^d} \psi_P(t) k(v-t) dt \tag{20}$$

**Lemma 3.** *[5] If $k$ is a continuous, integrable and translation invariant kernel with an inverse Fourier transform strictly greater then zero an and $P$ has integrable characteristic function, then the mapping:*

$$\Gamma : P \to \Phi_P \tag{21}$$

*is injective and $\Phi_P$ is an element of the RKHS $H_k$ associated with $k$.*

**Theorem 3.** *Given $p \geq 1$, $k$ a translation invariant with an inverse Fourier transform strictly greater then zero, continuous, and integrable kernel on $\mathbb{R}^d$, $\Phi_P$ and $\Phi_Q$ the smooth characteristic functions of the Borel probability measures with integrable characteristic functions $P$ and $Q$ respectively, the following function:*

$$d_{L^p,\Phi}(P,Q) := \left( \int_{t \in \mathbb{R}^d} \left| \Phi_P(t) - \Phi_Q(t) \right|^p d\Gamma(t) \right)^{1/p} \tag{22}$$

*where $\Gamma$ a Borel probability measure absolutely continuous with respect to Lebesgue measure, is a metric on the space of Borel probability measures with integrable characteristic functions. Moreover if a sequence $(\alpha_n)_{n \geq 0}$ of Borel probability measures with integrable characteristic functions converges weakly towards $\alpha$ then $d_{L^1,\mu}(\alpha_n, \alpha) \to 0$.*

**Proof.** *Let $p \geq 1$. First, as $\psi_P$ and $\psi_Q$ live in $H_k$, the RKHS associated with $k$, we have:*

$$|\Phi_P(t) - \Phi_Q(t)|^p \leq \|\Phi_P - \Phi_Q\|_H^p \, k(0)^{p/2}$$

*Let us prove now that if $P \neq Q$ then $d(P,Q) > 0$. Thanks to Lemma 1, $\Phi_P$ and $\Phi_Q$ are continuous. Since the mapping $P \to \Phi_P$ is injective, there must exists at least one point $o$ where $\Phi_P - \Phi_Q$ is non-zero. By continuity of $\Phi_P - \Phi_Q$, there exists a ball around $o$ in which $\Phi_P - \Phi_Q$ is non-zero. Then $d_{L^p,\Phi}(P,Q) > 0$. Moreover, all the other proprieties of a metric are clearly verified by this function. Let us now show that $d_{L^1,\Phi}$ admits a IPM formulation:*

**Lemma 4.** *Let $\mathcal{T}_k$ be the integral operator on $L_2^{d\Gamma}(\mathbb{R}^d)$ associated with the kernel $k$. By denoting $B_\infty^{d\Gamma}$ the unit ball of $L_\infty^{d\Gamma}(\mathbb{R}^d)$, we have that:*

$$d_{L^1,\Phi}(P,Q) = \sup_{f \in \mathcal{L}(\mathcal{T}_k(B_\infty^{d\Gamma}))} \left( \mathbb{E}_P[f(X)] - \mathbb{E}_Q[f(Y)] \right)$$

*where:*

$$\mathcal{L}(f)(x) := \int_{t \in \mathbb{R}^d} \exp(it^T x) f(t) dt \tag{23}$$

**Proof.** *Let $P$ and $Q$ be Borel probability measures with integrable characteristic functions. As $\Phi_P$ and $\Phi_Q$ live in the RKHS associated with $k$, we obtain, as in the proof of Theorem 2.1, that:*

$$d_{L^1,\Phi}(P,Q) = \langle \Phi_P - \Phi_Q, f \rangle$$

*with*

$$f = \int_{t \in \mathbb{R}^d} g_t d\Gamma(t) \quad where \quad g_t = \left\{ \begin{array}{l} k_t \; if \; t \in \{x : \Phi_P(x) \geq \Phi_Q(x)\} \\ -k_t \; otherwise. \end{array} \right.$$

*Therefore $f \in \mathcal{T}_k(B_\infty^{d\Gamma})$ and we have:*

$$d_{L^1,\Phi}(P,Q) = \int_{\mathbb{R}^d} \psi_P(t) f(t) dt - \int_{\mathbb{R}^d} \psi_Q(t) f(t) dt$$

$$= \int_{t \in \mathbb{R}^d} \int_{\epsilon \in \mathbb{R}^d} \exp(i\epsilon^T t) f(t) dP(\epsilon) dt - \int_{t \in \mathbb{R}^d} \int_{\epsilon \in \mathbb{R}^d} \exp(i\epsilon^T t) f(t) dQ(\epsilon) dt$$

*Let us now show that for any $g \in B_\infty^{d\Gamma}$, $\mathcal{T}_k(g)$ is integrable (w.r.t the Lebesgue measure):*

$$\int_{x \in \mathbb{R}^d} |\mathcal{T}_k(g)(x)| dx \leq \int_{x \in \mathbb{R}^d} \int_{t \in \mathbb{R}^d} |k(x,t) g(t)| d\Gamma(t) dx$$

*But as $k$ is translation-invariant we have:*

$$\int_{t \in \mathbb{R}^d} \int_{x \in \mathbb{R}^d} |k(x,t) g(t)| d\Gamma(t) dx = \int_{x \in \mathbb{R}^d} \left( \int_{u \in \mathbb{R}^d} |k(u)| du \right) |g(t)| d\Gamma(t)$$

$$= \int_{u \in \mathbb{R}^d} |k(u)| du \int_{x \in \mathbb{R}^d} |g(t)| d\Gamma(t)$$

*And as $k$ is integrable, and $g \in B_\infty^{d\Gamma}$, we can apply the Fubini–Tonelli theorem, and $\mathcal{T}_k(g)$ is integrable.*

*Therefore for any Borel probability measure $P$ with integrable characteristic function, $\int_{x \in \mathbb{R}^d} \int_{\epsilon \in \mathbb{R}^d} |f(t)| dP(\epsilon) dt < \infty$ and by Fubini–Tonelli theorem, we can rewrite $d_{L^1,\Phi}(P,Q)$ as:*

$$d_{L^1,\Phi}(P,Q) = \int_{\epsilon \in \mathbb{R}^d} \left( \int_{t \in \mathbb{R}^d} \exp(i\epsilon^T t) f(t) dt \right) dP(\epsilon) - \int_{\epsilon \in \mathbb{R}^d} \left( \int_{t \in \mathbb{R}^d} \exp(i\epsilon^T t) f(t) dt \right) dQ(\epsilon)$$

*Therefore we have:*

$$d_{L^1,\Phi}(P,Q) = \int_{\epsilon \in \mathbb{R}^d} \mathcal{L}(f)(\epsilon) dP(\epsilon) - \int_{\epsilon \in \mathbb{R}^d} \mathcal{L}(f)(\epsilon) dQ(\epsilon)$$

$$= \mathbb{E}_P(\mathcal{L}(f)(X)) - \mathbb{E}_Q(\mathcal{L}(f)(Y))$$

*Let now $g$ be an abritary function in $B_\infty^{d\Gamma}$. Then we have:*

$$\mathbb{E}_P(L(\mathcal{T}_k(g))(X)) - \mathbb{E}_Q(L(\mathcal{T}_k(g))(Y)) = \int_{\epsilon \in \mathbb{R}^d} \mathcal{L}(\mathcal{T}_k(g))(\epsilon) dP(\epsilon) - \int_{\epsilon \in \mathbb{R}^d} \mathcal{L}(\mathcal{T}_k(g))(\epsilon) dQ(\epsilon)$$

*But we have that:*

$$\int_{\epsilon \in \mathbb{R}^d} \mathcal{L}(\mathcal{T}_k(g))(\epsilon) dP(\epsilon) = \int_{\epsilon \in \mathbb{R}^d} \left( \int_{t \in \mathbb{R}^d} \exp(i\epsilon^T t) \mathcal{T}_k(g)(t) dt \right) dP(\epsilon)$$

$$= \int_{t \in \mathbb{R}^d} \left( \int_{\epsilon \in \mathbb{R}^d} \exp(i\epsilon^T t) dP(\epsilon) \right) \mathcal{T}_k(g)(t) dt$$

$$= \int_{\mathbb{R}^d} \psi_P(t) \mathcal{T}_k(g)(t) dt$$

$$= \langle \Phi_P, \mathcal{T}_k(g) \rangle$$

*Finally we have:*

$$\mathbb{E}_P(\mathcal{L}(\mathcal{T}_k(g))(X)) - \mathbb{E}_Q(\mathcal{L}(\mathcal{T}_k(g))(Y)) = \langle \Phi_P - \Phi_Q, \mathcal{T}_k(g) \rangle$$
$$= \int_{\mathbb{R}^d} g(t)(\Phi_P(t) - \Phi_Q(t)) d\Gamma(t)$$
$$\leq \int_{\mathbb{R}^d} |\Phi_P(t) - \Phi_Q(t)| d\Gamma(t)$$

*The results follows.*

*Therefore thanks to the IPM formulation of the $d_{L^1,\Phi}$-distance, we deduce that for all $p \geq 1$, if $\alpha_n$ converge weakly towards $\alpha$, then $d_{L^1,\Phi}(\alpha_n, \alpha) \to 0$. Indeed, we have shown that $\mathcal{T}_k(B_\infty^{d\Gamma}) \subset L^1(\mathbb{R}^d)$, therefore $\mathcal{L}(\mathcal{T}_k(B_\infty^{d\Gamma})) \subset C^0(\mathbb{R}^d)$, and the result follows.*

## B  Two-sample testing using the $\ell_1$ norm

### B.1  $\ell_1$-based random metric with mean embeddings

**Definition 3.** *Let k be a kernel. For any $J > 0$, we define:*

$$d_{\ell_1,\mu,J} := \left\{ d_{\ell_1,\mu,J}[P,Q] = \frac{1}{J} \sum_{j=1}^{J} |\mu_P(T_j) - \mu_Q(T_j)|: \quad P, Q \in \mathcal{M}_+^1(\mathbb{R}^d) \right\}$$

*with $\{T_j\}_{j=1}^{J}$ sampled independently from the distribution $\Gamma$.*

**Theorem 4.** *Let k be a bounded, analytic, and characteristic kernel. Then for any $J > 0$, $d_{\ell_1,\mu,J}$ is a random metric on the space of Borel probability measures.*

**Proof.** *To prove this theorem we have first to introduce the fact that analytic functions are 'well behaved'.*

**Lemma 5.** *Let $\mu$ be absolutely continuous measure on $\mathbb{R}^d$ (wrt. the Lebesgue measure). Non-zero, analytic function f can be zero at most at the set of measure 0, with respect to the measure $\mu$.*

**Proof.** *If f is zero at the set with a limit point then it is zero everywhere. Therefore f can be zero at most at a set A without a limit point, which by definition is a discrete set (distance between any two points in A is greater then some $\epsilon > 0$). Discrete sets have zero Lebesgue measure (as a countable union of points with zero measure). Since $\mu$ is absolutely continuous then $\mu(A)$ is zero as well.*

*Let us now show how to build a random metric based on the $\ell_1$ norm.*

**Lemma 6.** *Let $\Lambda$ be an injective mapping from the space of the Borel probability measures into a space of analytic functions on $\mathbb{R}^d$. Define*

$$d_{\Lambda,J}[P,Q] := \frac{1}{J} \sum_{j=1}^{J} |\Lambda P(T_j) - \Lambda Q(T_j)|$$

*with $\{T_j\}_{j=1}^{J}$ sampled independently from the distribution $\Gamma$.*

*Then $d_{\Lambda,J}$ is a random metric.*

**Proof.** *Let $\Lambda P$ and $\Lambda Q$ be images of measures P and Q respectively. We want to apply Lemma 5 to the analytic function $f = \Lambda P - \Lambda Q$, with the measure $\Gamma$, to see that if $P \neq Q$ then $f \neq 0$ a.s. To do so, we need to show that $P \neq Q$ implies that f is non-zero. Since mapping to $\Gamma$ is injective, there must exists at least one point o where f is non-zero. By continuity of f, there exists a ball around o in which f is non-zero.*

*We have shown that $P \neq Q$ implies f is almost everywhere non zero which in turn implies that $d_{\Lambda,J}(P,Q) > 0$ a.s. If $P = Q$ then $f = 0$ and $d_{\Lambda,J}(P,Q) = 0$.*

*By the construction $d_{\Lambda,J}$ is clearly symmetric and satisfies the triangle inequality.*

*Before proving the theorem we need to introduce a Lemma:*

**Lemma 7.** *[5] If k is a bounded, analytic kernel on $\mathbb{R}^d \times \mathbb{R}^d$, then all functions in the RKHS $H$ associated with this kernel are analytic.*

*Since k is characteristic the mapping $\Lambda : P \to \mu_P$ is injective. Since k is a bounded, analytic kernel on $\mathbb{R}^d \times \mathbb{R}^d$, the Lemma 7 guarantees that $\mu_P$ is analytic, hence the image of $\Lambda$ is a subset of analytic functions. Therefore, we can use Lemma 6 to see that $d_{\Lambda,J}[P,Q] = d_{\ell_1,\mu,J}[P,Q]$ is a random metric and this concludes the proof.*

## B.2 A first test with finite-sample control

Let us now build a statistic based on an estimation of the random metric introduced in eq.7. Let $X = \{x_1, ..., x_{N_1}\}$ and $Y = \{y_1, ..., y_{N_2}\} \subset \mathbb{R}^d$ i.i.d. two samples drawn respectively from the Borel probability measures $P$ and $Q$. From these samples we define their empirical mean embeddings $\mu_X$ and $\mu_Y$:

$$\mu_X(T) := \frac{1}{N_1} \sum_{i=1}^{N_1} k(x_i, T), \qquad \mu_Y(T) := \frac{1}{N_2} \sum_{i=1}^{N_2} k(y_i, T)$$

And we define:

$$\mathbf{S}_{N_1,N_2} := \Big( \mu_X(T_1) - \mu_Y(T_1), ..., \mu_X(T_J) - \mu_Y(T_J) \Big) \tag{24}$$

with $\{T_j\}_{j=1}^J$ sampled independently from the distribution $\Gamma$. Finally we define a first statistic:

$$d_{\ell_1,\mu,J}[X,Y] := \frac{1}{J} \|S_{N_1,N_2}\|_1 \tag{25}$$

We now derive a control of the statistic:

**Proposition 1.** *With K such that $\sup\limits_{x,y \in \mathbb{R}^d} |k(x,y)| \leq \frac{K}{2}$,*

$$\mathbb{P}_{X,Y} \left( \Big| d_{\ell_1,\mu,J}[X,Y] - d_{\ell_1,\mu,J}[P,Q] \Big| > t \right) \leq 2J \exp\left( \frac{-t^2 N_1 N_2}{2K^2(N_1 + N_2)} \right)$$

**Proof.** *We have:*

$$|d_{\ell_1,\mu,J}[X,Y] - d_{\ell_1,\mu,J}[P,Q]| \leq \frac{1}{J} \sum_{j=1}^J \Big| |\mu_X(T_j) - \mu_Y(T_j)| - |\mu_p(T_j) - \mu_p(T_j)| \Big|$$

*Then:*

$$|d_{\ell_1,\mu,J}[X,Y] - d_{\ell_1,\mu,J}[P,Q]| \leq \frac{1}{J} \sum_{j=1}^J \Big| (\mu_X(T_j) - \mu_Y(T_j)) - \mathbb{E}_{X,Y \sim p,q}(\mu_X(T_j) - \mu_Y(T_j)) \Big|$$

*Let us now consider the upper bound of the difference. By applying a union bound we have:*

$$\mathbb{P}\Big( \frac{1}{J} \sum_{j=1}^J \Big| (\mu_X(T_j) - \mu_Y(T_j)) - \mathbb{E}_{X,Y}(\mu_X(T_j) - \mu_Y(T_j)) \Big| \geq t \Big)$$

$$\leq \sum_{j=1}^J \mathbb{P}_{X,Y} \left( \frac{1}{J} \Big| (\mu_X(T_j) - \mu_Y(T_j)) - \mathbb{E}_{X,Y}(..) \Big| \geq \frac{t}{J} \right)$$

*Then by applying Hoeffding's inequality on each term of the sum of the right term of the inequality, we have:*

$$\mathbb{P}_{X,Y} \left( \frac{1}{J} \Big| (\mu_X(T_j) - \mu_Y(T_j)) - \mathbb{E}_{X,Y}(..) \Big| \geq \frac{t}{J} \right) \leq 2 \exp\left( -\frac{t^2 N_1 N_2}{2K^2(N_1 + N_2)} \right)$$

*Finally we have:*

$$\mathbb{P}_{X,Y} \left( \Big| d_{\ell_1,\mu,J}[X,Y] - d_{\ell_1,\mu,J}[P,Q] \Big| \geq t \right) \leq 2J \exp\left( -\frac{t^2 N_1 N_2}{2K^2(N_1 + N_2)} \right)$$

**Corollary 1.** *The hypothesis test associated with the statistic $d_{\ell_1,\mu,J}[X,Y]$ of level $\alpha$ for the null hypothesis $P = Q$, that is for $d_{\ell_1,\mu,J}[P,Q] = 0$ almost surely, has almost surely the acceptance region:*

$$d_{\ell_1,\mu,J}[X,Y] < K\sqrt{\frac{N_1 + N_2}{N_1 N_2}}\sqrt{2\log\left(\frac{J}{\alpha}\right)}$$

*Moreover, the test is consistent almost surely.*

*Proof.* Let us note the probability space of random variables $\{T_j\}_{j=1}^J$ as $(\Omega, \mathcal{F}, P)$.

Let $\omega \in \Omega$ such that $d^{\omega}_{\ell_1,\mu,J}[P,Q] = 0$. Then we have thanks to Proposition **??** that:

$$d^{\omega}_{\ell_1,\mu,J}[X,Y] < K\sqrt{\frac{N_1 + N_2}{N_1 N_2}}\sqrt{2\log\left(\frac{J}{\alpha}\right)}$$

with a probability at last of $1 - \alpha$.

By assuming the null hypothesis $P = Q$, we have thanks to Theorem 4 that $d_{\ell_1,\mu,J}[P,Q] = 0$ a.s., then the result above hold a.s.

Moreover the statistic converges in probability to its population value a.s which give us the consistency of the test a.s. $\qquad\square$

We now show that, under the alternative hypothesis, the statistic captures dense differences between distributions with high probability:

**Corollary 2.** *Let $\gamma > 0$, then under the alternative hypothesis, almost surely there exist $\Delta > 0$ such that for all $N_1, N_2 \geq 1$:*

$$\mathbb{P}_{X,Y}\left(\forall j \in [\![1, J]\!], \frac{|\mu_X(T_j) - \mu_Y(T_j)|}{J} \geq \frac{\Delta}{J} - \omega_{N_1,N_2}\right) \geq 1 - \gamma$$

$$\text{where } \omega_{N_1,N_2} = \frac{1}{J}\sqrt{\log(\frac{J^2}{\gamma})\frac{2K^2(N_1 + N_2)}{N_1 N_2}}$$

*Proof.* Let $\Delta$ be the minimum of $\mu_p - \mu_q$ over the set of locations $\{T_j\}_{j=1}^J$. Thanks to the analycity of the kernel we have that under the alternative hypothesis, $\mu_p - \mu_q$ is non zero everywhere almost surely. Therefore $\Delta > 0$ almost surely. Moreover by applying Proposition 1 for each $T_j$ we obtain that for all $N_1, N_2 \geq 0$:

$$\mathbb{P}_{X,Y}\left(\frac{|\mu_X(T_j) - \mu_Y(T_j)|}{J} \geq \frac{\Delta}{J} - \omega_{N_1,N_2}\right) \geq 1 - \frac{\gamma}{J}$$

$$\text{where } \omega_{N_1,N_2} = \frac{1}{J}\sqrt{\log(\frac{J^2}{\gamma})\frac{2K^2(N_1 + N_2)}{N_1 N_2}}$$

Finally by applying an union bound, the result follows. $\qquad\square$

## C   A test statistic with simple asymptotic distribution

### C.1   Asymptotic distribution of $\widehat{d}_{\ell_1,\mu,J}[X,Y]$

**Proposition 2.** *Let $\{T_j\}_{j=1}^J$ sampled independently from the distribution $\Gamma$ and $X := \{x_i\}_{i=1}^n$ and $Y := \{y_i\}_{i=1}^n$ be i.i.d. samples from $P$ and $Q$ respectively. Under $H_0$, the statistic $\widehat{d}_{\ell_1,\mu,J}[X,Y]$ is almost surely asymptotically distributed as a sum of $J$ correlated Nakagami variables. Finally under $H_1$, almost surely the statistic can be arbitrarily large as $n \to \infty$, allowing the test to correctly reject $H_0$.*

**Proof.** *Let us note the probability space of random variables $\{T_j\}_{j=1}^J$ as $(\Omega, \mathcal{F}, P)$. Let $\omega \in \Omega$ such that $d^\omega_{\ell_1, \mu, J}[P, Q] = 0$ (see Definition 3) and let us define:*

$$\mathbf{z}_i^\omega := (k(x_i, T_1(\omega)) - k(y_j, T_1(\omega)), ..., k(x_i, T_J(\omega)) - k(y_j, T_J(\omega))) \in \mathbb{R}^J$$

*Therefore we can define:*

$$\mathbf{S}_n := \frac{1}{n} \sum_{i=1}^n \mathbf{z}_i^\omega$$

*By applying the Central-Limit Thoerem, we have:*

$$\sqrt{n}\mathbf{S}_n \to \mathcal{N}(0, \boldsymbol{\Sigma}^\omega) \qquad with \qquad \boldsymbol{\Sigma}^\omega := Cov(\mathbf{z}^\omega)$$

*Therefore $\widehat{d}^\omega_{\ell_1, \mu, J}[X, Y] = \|\sqrt{n}\,\mathbf{S}_n^\omega\|_1$ converges to a sum of correlated Nakagami variables. But under, the null hypothesis $P = Q$, we have thanks to Theorem 4 that $d_{\ell_1, \mu, J}[P, Q] = 0$ a.s., then a.s. $\widehat{d}_{\ell_1, \mu, J}$ converges to a sum of correlated Nakagami variables. Let's now consider an $\omega$ such that $d^\omega_{\ell_1, \mu, J}[P, Q] > 0$. Since $\mathbf{S}_n^\omega$ converges in probability to the vector $\mathbf{S}^\omega = \mathbb{E}(\mathbf{z}^\omega) \neq 0$, then we have:*

$$\mathbb{P}\left(\left\|\sqrt{n}\mathbf{S}_n^\omega\right\|_1 > r\right) = \mathbb{P}\left(\|\mathbf{S}_n^\omega\|_1 - \frac{r}{\sqrt{n}} > 0\right)$$

*And as $\frac{r}{\sqrt{t}} \to 0$ as $t \to \infty$, we have finally:*

$$\mathbb{P}\left(\left\|\sqrt{n}\mathbf{S}_n^\omega\right\|_1 > r\right) \to 1 \quad as \quad t \to \infty.$$

*Finally, under $H_1$, $d_{\ell_1, \mu, J}[P, Q] > 0$ almost surely and the statistic can be arbitrarily large as $n \to \infty$ almost surely.*

## C.2 Proof of Proposition 3.1

**Proposition 3.** *Let $\alpha \in ]0, 1[$, $\gamma > 0$ and $J \geq 2$. Let $\{T_j\}_{j=1}^J$ sampled i.i.d. from the distribution $\Gamma$ and let $X := \{x_i\}_{i=1}^n$ and $Y := \{y_i\}_{i=1}^n$ i.i.d. samples from $P$ and $Q$ respectively. Let us denote $\delta$ the $(1 - \alpha)$-quantile of the asymptotic null distribution of $\widehat{d}_{\ell_1, \mu, J}[X, Y]$ and $\beta$ the $(1 - \alpha)$-quantile of the asymptotic null distribution of $\widehat{d}^2_{\ell_2, \mu, J}[X, Y]$. Under the alternative hypothesis, almost surely, there exists $N \geq 1$ such that for all $n \geq N$, with a probability of at least $1 - \gamma$ we have:*

$$\widehat{d}^2_{\ell_2, \mu, J}[X, Y] > \beta \Rightarrow \widehat{d}_{\ell_1, \mu, J}[X, Y] > \delta \tag{26}$$

**Proof.** *First we remarks that:*

$$\widehat{d}^2_{\ell_2, \mu, J}[X, Y] = \|\sqrt{n}\mathbf{S}_n\|_2^2$$

*and*

$$\widehat{d}_{\ell_1, \mu, J}[X, Y] = \|\sqrt{n}\mathbf{S}_n\|_1$$

*where $\mathbf{S}_n := \frac{1}{n}\sum_{i=1}^n \mathbf{z}_i^\omega$ and $\mathbf{z}_i := (k(x_i, T_1(\omega)) - k(y_j, T_1(\omega)), ..., k(x_i, T_J(\omega)) - k(y_j, T_J(\omega)))$. Let us now introduce the following Lemma:*

**Lemma 8.** *Let $\mathbf{x}$ a random vector $\in \mathbb{R}^J$ with $J \geq 2$, $\mathbf{z} := \min_{j \in [[1, J]]} |x_j|$, $\epsilon > 0$ and $\gamma > 0$. If*

$$\mathbb{P}(\mathbf{z} \geq \epsilon) \geq 1 - \gamma$$

*we have with a probability of at least $1 - \gamma$ that, $\forall t_1 \geq t_2 \geq 0$, if $\epsilon \geq \sqrt{\frac{t_1^2 - t_2^2}{J(J-1)}}$, then*

$$\|\mathbf{x}\|_2 > t_2 \Rightarrow \|\mathbf{x}\|_1 > t_1.$$

**Proof.** *First we remarks that:*

$$\epsilon > \sqrt{\frac{t_1^2 - t_2^2}{J(J-1)}} \Rightarrow J(J-1)\epsilon > t_1^2 - t_2^2$$

$$\Rightarrow t_2^2 > t_1^2 - J(J-1)\epsilon^2$$

*Therefore, we have:*

$$\|\mathbf{x}\|_2 \geq t_2 \Rightarrow \|\mathbf{x}\|_2^2 + J(J-1)\epsilon^2 \geq t_1^2$$
$$\Rightarrow \sqrt{\|\mathbf{x}\|_2^2 + J(J-1)\epsilon^2} \geq t_1$$

*But we have that:*

$$\|\mathbf{x}\|_1^2 = \sum_{i=1}^{J} |\mathbf{x}_i|^2 + \sum_{i \neq j} |\mathbf{x}_i||\mathbf{x}_j|$$

*Therefore we have with a probability of 1-$\gamma$ that:*

$$\|\mathbf{x}\|_1^2 \geq \|\mathbf{x}\|_2^2 + J(J-1)\epsilon^2$$

*And:*

$$\|\mathbf{x}\|_2 \geq t_2 \Rightarrow \|\mathbf{x}\|_1 \geq t_1$$

*Moreover by denoting $\delta$ the $(1-\alpha)$-quantile of the asymptotic null distribution of $\widehat{d}_{\ell_1,\mu,J}[X,Y]$ and $\beta$ the $(1-\alpha)$-quantile of the asymptotic null distribution of $\widehat{d}_{\ell_2,\mu,J}^2[X,Y]$ we have that $\delta \geq \sqrt{\beta}$:*

**Lemma 9.** *Let $\mathbf{x}$ be a random vector in $\mathbb{R}^J$, $\delta$ the $(1-\alpha)$-quantile of $\|\mathbf{x}\|_1$ and $\beta$ the $(1-\alpha)$-quantile of $\|\mathbf{x}\|_2$. We have then:*

$$\delta \geq \beta \geq 0. \tag{27}$$

**Proof.** *The results is a direct consequence of the domination of the $\ell_1$ norm:*

$$\|\mathbf{x}\|_1 \geq \|\mathbf{x}\|_2$$

*Indeed, under $H_0$, we have shown that (see proof Proposition 2):*

$$\sqrt{n}\mathbf{S}_n \rightarrow \mathcal{N}(0,\boldsymbol{\Sigma}^\omega) \qquad \text{with} \qquad \boldsymbol{\Sigma} := Cov(\mathbf{z})$$

*Therefore by applying the Lemma 9 to $\mathbf{x}$ which follows $\mathcal{N}(0,\boldsymbol{\Sigma}^\omega)$, we obtain that $\delta \geq \sqrt{\beta}$. Now, To show the result we only need to show that the assumption of the Lemma 8 is sastified for the random vector $\mathbf{x} := \sqrt{n}\mathbf{S}_n$, $t_1 = \delta$ and $t_2 = \sqrt{\beta}$, i.e. for $\epsilon = \sqrt{\frac{\delta^2 - \beta}{J(J-1)}}$ under the alternative hypothesis. Under $H_1 : P \neq Q$, we have that $\mathbf{S}_n$ converge in probability to $\mathbf{S} := \mathbb{E}_{(x,y)\sim(P,Q)}(\mathbf{S}_n)$. Then by continuity of the application:*

$$\phi_j : x := (x_j)_{j=1}^{J} \mathbb{R}^J \rightarrow |x_j|$$

*, we have that for all $j \in [1,J]$, $|(\mathbf{S}_n)_j|$ converges in probability towards $\mathbf{S}_j$, the $j$-th coordinate of $\mathbf{S}$. Since $\mathbf{S} = (\mu_P(T_j) - \mu_Q(T_j))_{j=1}^{J}$, thanks to the analycity of the kernel $k$, the Lemma 7 guarantees the analycity of $\mu_P - \mu_Q$. And thanks to the injectivity of the mean embedding function, $\mu_P - \mu_Q$ is a non-zero function, therefore thanks to Lemma 5 $\mu_P - \mu_Q$ is non zero almost everywhere. Moreover the $(T_j)_{j=1}^{J}$ are independent, therefore the coordinates of $\mathbf{S}$ are almost surely all nonzero. Then we have then for all $j \in [\![1,J]\!]$:*

$$\mathbb{P}\left(\left|(\sqrt{n}\mathbf{S}_n)_j\right| > \epsilon\right) = \mathbb{P}\left(\left|(\mathbf{S}_n)_j\right| - \frac{\epsilon}{\sqrt{n}} > 0\right)$$

*And as $\frac{\epsilon}{\sqrt{n}} \rightarrow 0$ as $n \rightarrow \infty$, we have finally almost surely for all $j \in [\![1,J]\!]$:*

$$\mathbb{P}_{X,Y}\left(\left|(\sqrt{n}\mathbf{S}_n)_j\right| \geq \epsilon\right) \rightarrow 1 \text{ as } n \rightarrow \infty$$

*Therefore almost surely there exist $N \geq 1$ such that for all $n \geq N$ and for all $j \in [\![1,J]\!]$:*

$$\mathbb{P}_{X,Y}\left(\left|(\sqrt{n}\mathbf{S}_n)_j\right| \geq \epsilon\right) \geq 1 - \frac{\gamma}{J}$$

*Finally by applying a union bound we obtain that almost surely, for all $n \geq N$:*

$$\mathbb{P}_{X,Y}\left(\forall j \in [1,J], \left|(\sqrt{n}\mathbf{S}_n)_j\right| \geq \epsilon\right) \geq 1 - \gamma$$

*Therefore by applying Lemma 8, we obtain that, almost surely, for all $n \geq N$, with a probability of at least $1 - \gamma$:*

$$\|\sqrt{n}\mathbf{S}_n\|_2 > \sqrt{\beta} \Rightarrow \|\sqrt{n}\mathbf{S}_n\|_1 > \delta.$$

## C.3 Proof of the Proposition 3.2

**Proposition 4.** *Let $\{T_j\}_{j=1}^J$ sampled independently from the distribution $\Gamma$ and $X := \{x_i\}_{i=1}^{N_1}$ and $Y := \{y_i\}_{i=1}^{N_2}$ be i.i.d. samples from $P$ and $Q$ respectively. Under $H_0$, the statistic L1-ME$[X,Y]$ is almost surely asymptotically distributed as $Naka(\frac{1}{2}, 1, J)$, a sum of $J$ random variables i.i.d which follow a Nakagami distribution of parameter $m = \frac{1}{2}$ and $\omega = 1$. Finally under $H_1$, almost surely the statistic can be arbitrarily large as $t \to \infty$, allowing the test to correctly reject $H_0$.*

**Proof.** *Let us note the probability space of random variables $\{T_j\}_{j=1}^J$ as $(\Omega, \mathcal{F}, P)$. Let $\omega \in \Omega$ such that $d_{\ell_1,\mu,J}^\omega[P,Q] = 0$ (see Definition 3). Let us denote:*

$$\mathbf{Z}_X^{i,\omega} := (k(x_i, T_1(\omega)), ..., k(x_i, T_J(\omega))) \qquad \mathbf{Z}_Y^{j,\omega} := (k(y_j, T_1(\omega)), ..., k(y_j, T_J(\omega))),$$

$$\mathbf{S}_{N_1,N_2}^\omega := \frac{1}{N_1} \sum_{i=1}^{N_1} \mathbf{Z}_X^{i,\omega} - \frac{1}{N_2} \sum_{j=1}^{N_2} \mathbf{Z}_Y^{j,\omega}.$$

*As $d_{\ell_1,\mu,J}^\omega[P,Q] = 0$ then for all $j$, $\mu_p(T_j(\omega)) = \mu_q(T_j(\omega))$, which implies that $\mathbb{E}\left(\mathbf{Z}_X^{i,\omega}\right) = \mathbb{E}\left(\mathbf{Z}_Y^{j,\omega}\right)$. Therefore, by applying the Central-Limit Theorem, we have:*

$$\sqrt{t}\, \mathbf{S}_{N_1,N_2}^\omega \longrightarrow \mathcal{N}\left(0, \frac{\mathbf{\Sigma}_1^\omega}{\rho}\right) - \mathcal{N}\left(0, \frac{\mathbf{\Sigma}_2^\omega}{1-\rho}\right) \qquad with \quad \mathbf{\Sigma}_1 = Cov(\mathbf{Z}_X^\omega) \quad and \quad \mathbf{\Sigma}_2 = Cov(\mathbf{Z}_Y^\omega)$$

*As $\mathbf{Z}_X^\omega$ and $\mathbf{Z}_Y^\omega$ are independent, we have then that:*

$$\sqrt{t}\, \mathbf{S}_{N_1,N_2}^\omega \longrightarrow \mathcal{N}(0, \mathbf{\Sigma}^\omega) \qquad with \quad \mathbf{\Sigma}^\omega = \frac{\mathbf{\Sigma}_1^\omega}{\rho} + \frac{\mathbf{\Sigma}_2^\omega}{1-\rho}$$

*And by Slutsky's theorem we deduce that:*

$$\sqrt{t}\left(\mathbf{\Sigma}_{N_1,N_2}^\omega\right)^{-\frac{1}{2}} \mathbf{S}_{N_1,N_2}^\omega \longrightarrow \mathcal{N}(0, \mathbf{I})$$

*So by noting, $\sqrt{t}\left(\mathbf{\Sigma}_{N_1,N_2}^\omega\right)^{-\frac{1}{2}} \mathbf{S}_{N_1,N_2}^\omega = \left(W_{N_1,N_2}^{1,\omega}, ..., W_{N_1,N_2}^{J,\omega}\right)$, we have that for each coordinate:*

$$\left(W_{N_1,N_2}^{j,\omega}\right) \longrightarrow \mathbf{S}_j^\omega$$

*where $\left(\mathbf{S}_j^\omega\right)$ are i.i.d and follow a standard normal distribution. Therefore by considering the $\ell_1$ norm of the statistic we have that:*

$$||\sqrt{t}\left(\mathbf{\Sigma}_{N_1,N_2}^\omega\right)^{-\frac{1}{2}} \mathbf{S}_{N_1,N_2}^\omega||_1 \longrightarrow \sum_{j=1}^J |\mathbf{S}_j^\omega|$$

*where $\left(\mathbf{S}_j^\omega\right)$ are independent and $\mathbf{S}_j^\omega \sim Naka\left(\frac{1}{2}, 1\right)$. And by assuming the null hypothesis $P = Q$, we have thanks to Theorem 4 that $d_{\ell_1,\mu,J}[P,Q] = 0$ a.s., then the result above hold a.s. Moreover, let's consider an $\omega$ such that $d_{\ell_1,\mu,J}^\omega[P,Q] > 0$. First we need show that $\left(\mathbf{\Sigma}_{N_1,N_2}^\omega\right)^{-\frac{1}{2}}$ converges in probability to the positive definite matrix $(\mathbf{\Sigma}^\omega)^{-\frac{1}{2}}$. For that we need to prove the following:*

**Lemma 10.** *The function $h(\mathbf{X}) = \mathbf{X}^{-\frac{1}{2}}$ is well defined on $\mathcal{S}_J^{++}(R)$ and is continuous.*

**Proof.** *First we observe that $h$ is the composition of two function which are:*

- *$h_1(\mathbf{X}) = \mathbf{X}^{-1}$ which is well defined and continuous on $\mathcal{S}_J^{++}(R)$*

- *$h_2(\mathbf{X}) = \mathbf{X}^{\frac{1}{2}}$ which is well defined on $\mathcal{S}_J^+(R)$ because each matrix of $\mathcal{S}_J^+(R)$ admits a unique square root matrix on $\mathcal{S}_J^+(R)$, so the result hold on $\mathcal{S}_J^{++}(R)$.*

*Let us prove now the continuity of $h_2$. Let $(\mathbf{U}_n)$ a sequence in $\mathcal{S}_n^{++}(R)$ such that $\mathbf{U}_n \to \mathbf{U}$ and let us prove that $h_2(\mathbf{U}_n) \to h_2(\mathbf{U})$ to prove the continuity of $h_2$. As $(\mathbf{U}_n)$ converges, then $(\mathbf{U}_n)$ is bounded, and we have:*

$$|||\mathbf{U}_n||| \leq K \implies |||h_2(\mathbf{U}_n)||| = \sqrt{|||\mathbf{U}_n|||} \leq \sqrt{K}$$

Then $(h_2(\mathbf{U}_n))$ is bounded. Let us show now that: $\forall \mathbf{A}$ s.t $\exists \phi$ strictly increasing and $h_2(\mathbf{U}_{\phi(n)}) \to \mathbf{A}$ we have $\mathbf{A} = h_2(\mathbf{U})$. Let $\mathbf{A}$ defined as above. Then $\exists \phi$ strictly increasing such that $h_2(\mathbf{U}_{\phi(n)}) \to \mathbf{A}$. As $\mathcal{S}_n^+(R)$ is closed, $\mathbf{A} \in \mathcal{S}_n^+(R)$, and by continuity of $\mathbf{M} \to \mathbf{M}^2$ we have also that $\mathbf{U}_{\phi(n)} \to \mathbf{A}^2$. And as $\mathbf{U}_n \to \mathbf{U}$, we have $\mathbf{A}^2 = \mathbf{U}$. And by uniqueness, we have finally:

$$h_2(\mathbf{U}) = \mathbf{A}.$$

So $h_2$ est continuous, and that conclude the proof.

Then each entry of the matrix $\boldsymbol{\Sigma}_{N_1,N_2}^\omega$ converges to the matrix $\boldsymbol{\Sigma}^\omega$, hence entires of the matrix $(\boldsymbol{\Sigma}^\omega)^{-\frac{1}{2}}$, given by a continuous function of the entries of $\boldsymbol{\Sigma}^\omega$, are limit of the sequence $(\boldsymbol{\Sigma}_{N_1,N_2}^\omega)^{-\frac{1}{2}}$.

Similarly $\mathbf{S}_{N_1,N_2}^\omega$ converges in probability to the vector $\mathbf{S}^\omega = \mathbb{E}(\mathbf{Z}^{1,\omega}) - \mathbb{E}(\mathbf{Z}^{2,\omega}) \neq 0$. Since $\|(\boldsymbol{\Sigma}^\omega)^{-\frac{1}{2}}\mathbf{S}^\omega\|_1 = \mathbf{A}_\omega > 0$ (indeed $(\boldsymbol{\Sigma}^\omega)^{-\frac{1}{2}}$ is positive definite), then $\|(\boldsymbol{\Sigma}_{N_1,N_2}^\omega)^{-\frac{1}{2}}\mathbf{S}_{N_1,N_2}^\omega\|_1$, being a continuous function of the entries of $\mathbf{S}_{N_1,N_2}^\omega$ and $(\boldsymbol{\Sigma}_{N_1,N_2}^\omega)^{-\frac{1}{2}}$, converges to $\mathbf{A}_\omega$. Then

$$\mathbb{P}\left(\left\|\sqrt{t}(\boldsymbol{\Sigma}_{N_1,N_2}^\omega)^{-\frac{1}{2}}\mathbf{S}_{N_1,N_2}^\omega\right\|_1 > r\right) = \mathbb{P}\left(\left\|(\boldsymbol{\Sigma}_{N_1,N_2}^\omega)^{-\frac{1}{2}}\mathbf{S}_{N_1,N_2}^\omega\right\|_1 - \frac{r}{\sqrt{t}} > 0\right)$$

And as $\frac{r}{\sqrt{t}} \to 0$ as $t \to \infty$, we have finally:

$$\mathbb{P}\left(\left\|\sqrt{t}\left(\boldsymbol{\Sigma}_{N_1,N_2}^\omega\right)^{-\frac{1}{2}}\mathbf{S}_{N_1,N_2}^\omega\right\|_1 > r\right) \to 1 \quad as \quad t \to \infty.$$

Finally, since $d_{\ell_1,\mu,J}[P,Q] > 0$ almost surely then $\mathbb{E}(\mathbf{Z}^{1,\omega}) - \mathbb{E}(\mathbf{Z}^{2,\omega}) \neq 0$ for almost all $\omega \in \Omega_1$, therefore under $H_1$, the statistic can be arbitrarily large as $t \to \infty$ almost surely.

## D  Optimizing test locations to improve power

### D.1  Proof of Proposition 3.3

**Proposition D.1.** Let $\mathcal{K}$ be a uniformly bounded family of $k : \mathbb{R}^d \times \mathbb{R}^d \to \mathbb{R}$ measurable kernels (i.e., $\exists K < \infty$ such that $\sup_{k \in \mathcal{K}} \sup_{(x,y) \in (\mathbb{R}^d)^2} |k(x,y)| \leq K$). Let $\mathcal{V}$ be a collection in which each element is a set of $J$ test locations. Assume that $c := \sup_{V \in \mathcal{V}, k \in \mathcal{K}} \|\boldsymbol{\Sigma}^{-1/2}\| < \infty$. Then the test power $\mathbb{P}\left(\widehat{\lambda}_t \geq \delta\right)$ of the L1-ME test satisfies $\mathbb{P}\left(\widehat{\lambda}_t \geq \delta\right) \geq L(\lambda_t)$ where:

$$L(\lambda_t) = 1 - 2\sum_{k=1}^J \exp\left(-\left(\frac{\lambda_t - \delta}{J^2 + J}\right)^2 \frac{\gamma_{N_1,N_2} N_1 N_2}{(N_1 + N_2)^2}\right)$$

$$- 2\sum_{k,q=1}^J \exp\left(-2\frac{\left(\frac{\gamma_{N_1,N_2}}{K_3 J^2}\frac{\lambda_t - \delta}{(J^2+J)\sqrt{t}} - \frac{J^3 K_2}{\sqrt{\gamma_{N_1,N_2}}} - J^4 K_1\right)^2}{K_\lambda^2 (N_1 + N_2)\max\left(\frac{8}{\rho N_1}, \frac{8}{(1-\rho)N_2}\right)^2}\right)$$

and $K_1, K_2, K_3$ and $K_\lambda$, are positive constants depending on only $K$, $J$ and $c$. The parameter $\lambda_t := \|\sqrt{t}\boldsymbol{\Sigma}^{-\frac{1}{2}}\mathbf{S}\|_1$ is the population counterpart of $\widehat{\lambda}_t := \|\sqrt{t}(\boldsymbol{\Sigma}_{N_1,N_2} + \gamma_{N_1,N_2}\mathbf{I})^{-\frac{1}{2}}\mathbf{S}_{N_1,N_2}\|_1$ where $\mathbf{S} = \mathbb{E}_{x,y}(S_{N_1,N_2})$ and $\boldsymbol{\Sigma} = \mathbb{E}_{x,y}(\boldsymbol{\Sigma}_{N_1,N_2})$. Moreover for large $t$, $L(\lambda_t)$ is increasing in $\lambda_t$.

**Proof.** We will first find an upper bound of $|\widehat{\lambda}_t - \lambda_t|$, then we will compute a lower bound of $\mathbb{P}\left(\widehat{\lambda}_t > \delta\right)$. To simplify the notation In the following, we denote:

$$\boldsymbol{\Sigma}_{N_1,N_2} := \frac{\boldsymbol{\Sigma}_{N_1}}{\rho} + \frac{\boldsymbol{\Sigma}_{N_2}}{1 - \rho} + \gamma_{N_1,N_2}\mathbf{I} \tag{28}$$

such that $\widehat{\lambda}_t := \|\sqrt{t}(\boldsymbol{\Sigma}_{N_1,N_2})^{-\frac{1}{2}}\mathbf{S}_{N_1,N_2}\|_1$. We have:

$$|\widehat{\lambda}_{N_1,N_2} - \lambda_t| = \left|\sqrt{t}\left(\|\boldsymbol{\Sigma}_{N_1,N_2}^{-\frac{1}{2}}\mathbf{S}_{N_1,N_2}\|_1 - \|\boldsymbol{\Sigma}^{-\frac{1}{2}}\mathbf{S}\|_1\right)\right|$$

*Then we have:*

$$\left| \|\boldsymbol{\Sigma}_{N_1,N_2}^{-\frac{1}{2}}\mathbf{S}_{N_1,N_2}\|_1 - \|\boldsymbol{\Sigma}^{-\frac{1}{2}}\mathbf{S}\|_1 \right| \leq \|\boldsymbol{\Sigma}_{N_1,N_2}^{-\frac{1}{2}}\mathbf{S}_{N_1,N_2} - \boldsymbol{\Sigma}^{-\frac{1}{2}}\mathbf{S}\|_1$$

$$\leq \|\boldsymbol{\Sigma}_{N_1,N_2}^{-\frac{1}{2}}\mathbf{S}_{N_1,N_2} - \boldsymbol{\Sigma}_{N_1,N_2}^{-\frac{1}{2}}\mathbf{S} + \boldsymbol{\Sigma}_{N_1,N_2}^{-\frac{1}{2}}\mathbf{S} - \boldsymbol{\Sigma}^{-\frac{1}{2}}\mathbf{S}\|_1$$

$$\leq \|\boldsymbol{\Sigma}_{N_1,N_2}^{-\frac{1}{2}}\left(\mathbf{S}_{N_1,N_2} - \mathbf{S}\right)\|_1 + \|\left(\boldsymbol{\Sigma}_{N_1,N_2}^{-\frac{1}{2}} - \boldsymbol{\Sigma}^{-\frac{1}{2}}\right)\mathbf{S}\|_1$$

*Let us now consider the first term on the right side of the inequality:*

$$\|\boldsymbol{\Sigma}_{N_1,N_2}^{-\frac{1}{2}}\left(\mathbf{S}_{N_1,N_2} - \mathbf{S}\right)\|_1 = \sum_{j=1}^{J}|\boldsymbol{\Sigma}_{N_1,N_2}^{-\frac{1}{2}}\left(\mathbf{S}_{N_1,N_2} - \mathbf{S}\right)|_j$$

*But since $\boldsymbol{\Sigma}_{N_1,N_2}$ is symmetric definite positive, we can write:*

$$\boldsymbol{\Sigma}_{N_1,N_2} = \mathbf{U}\mathbf{D}\mathbf{U}^T$$

*where $\mathbf{U}$ is orthogonal and $\mathbf{D} = diag\left(\lambda_i\right)$ with $\lambda_i > 0$. So:*

$$\boldsymbol{\Sigma}_{N_1,N_2}^{-\frac{1}{2}} = \mathbf{U}\mathbf{D}^{-\frac{1}{2}}\mathbf{U}^T$$

*But the regularization of $\boldsymbol{\Sigma}_{N_1,N_2} = (\frac{\boldsymbol{\Sigma}_{N_1}}{\rho} + \frac{\boldsymbol{\Sigma}_{N_2}}{1-\rho} + \gamma_{N_1,N_2}\mathbf{I})$ ensure that $\lambda_i \geq \gamma_{N_1,N_2}$. Thus $\lambda_i^{-\frac{1}{2}} \leq \gamma_{N_1,N_2}^{-\frac{1}{2}}$, and we have now:*

$$\left|[\boldsymbol{\Sigma}_{N_1,N_2}^{-\frac{1}{2}}]_{i,j}\right| = \left|\sum_{j=1}^{J}\lambda_j^{-\frac{1}{2}}\left(\mathbf{U}_k\right)_i\left(\mathbf{U}_k\right)_j\right|$$

*where $\mathbf{U} = [\mathbf{U}_1, ..., \mathbf{U}_J]$ and $\|\mathbf{U}_k\|_2 = 1$. And finally:*

$$\left|[\boldsymbol{\Sigma}_{N_1,N_2}^{-\frac{1}{2}}]_{i,j}\right| \leq \frac{J}{\sqrt{\gamma_{N_1,N_2}}}$$

*Now we have:*

$$\left\|\boldsymbol{\Sigma}_{N_1,N_2}^{-\frac{1}{2}}\left(\mathbf{S}_{N_1,N_2} - \mathbf{S}\right)\right\|_1 \leq \sum_{j=1}^{J}\left|\sum_{k=1}^{J}[\boldsymbol{\Sigma}_{N_1,N_2}^{-\frac{1}{2}}]_{j,k}\left(\mathbf{S}_{N_1,N_2} - \mathbf{S}\right)_k\right|$$

$$\leq \frac{J^2}{\sqrt{\gamma_{N_1,N_2}}}\sum_{k=1}^{J}|\left(\mathbf{S}_{N_1,N_2} - \mathbf{S}\right)_k|$$

$$\leq \frac{J^2}{\sqrt{\gamma_{N_1,N_2}}}\sum_{k=1}^{J}|\mu_X\left(T_k\right) - \mu_Y\left(T_k\right) - \mathbb{E}\left(\mu_X\left(T_k\right) - \mu_Y\left(T_k\right)\right)|$$

*Let us note $\frac{\boldsymbol{\Sigma}_{N_1}}{\rho} + \frac{\boldsymbol{\Sigma}_{N_2}}{1-\rho} = \mathbf{M}_{N_1,N_2}$ and consider the second term of the inequality:*

$$\boldsymbol{\Sigma}_{N_1,N_2}^{-\frac{1}{2}} - \boldsymbol{\Sigma}^{-\frac{1}{2}} = (\mathbf{M}_{N_1,N_2} + \gamma_{N_1,N_2}\mathbf{I})^{-\frac{1}{2}} - \boldsymbol{\Sigma}^{-\frac{1}{2}}$$

$$= \left[(\mathbf{M}_{N_1,N_2} + \gamma_{N_1,N_2}\mathbf{I})^{-\frac{1}{2}} - (\boldsymbol{\Sigma} + \gamma_{N_1,N_2}\mathbf{I})^{-\frac{1}{2}}\right] + \left[(\boldsymbol{\Sigma} + \gamma_{N_1,N_2}\mathbf{I})^{-\frac{1}{2}} - \boldsymbol{\Sigma}\right]$$

$$= (1) + (2)$$

*Let us first consider $(1)$:*

$$(1) = \boldsymbol{\Sigma}_{N_1,N_2}^{-\frac{1}{2}}\left((\boldsymbol{\Sigma} + \gamma_{N_1,N_2}\mathbf{I})^{\frac{1}{2}} - (\mathbf{M}_{N_1,N_2} + \gamma_{N_1,N_2}\mathbf{I})^{\frac{1}{2}}\right)(\boldsymbol{\Sigma} + \gamma_{N_1,N_2}\mathbf{I})^{-\frac{1}{2}}$$

$$= \boldsymbol{\Sigma}_{N_1,N_2}^{-\frac{1}{2}}\left[(\mathbb{E}\left(\mathbf{M}_{N_1,N_2} + \gamma_{N_1,N_2}\mathbf{I}\right))^{\frac{1}{2}} - (\mathbf{M}_{N_1,N_2} + \gamma_{N_1,N_2}\mathbf{I})^{\frac{1}{2}}\right](\mathbb{E}\left(\mathbf{M}_{N_1,N_2} + \gamma_{N_1,N_2}\mathbf{I}\right))^{\frac{1}{2}}$$

$$= \boldsymbol{\Sigma}_{N_1,N_2}^{-\frac{1}{2}}\left[(\mathbb{E}\left(\boldsymbol{\Sigma}_{N_1,N_2}\right))^{\frac{1}{2}} - \boldsymbol{\Sigma}_{N_1,N_2}^{\frac{1}{2}}\right](\mathbb{E}\left(\boldsymbol{\Sigma}_{N_1,N_2}\right))^{-\frac{1}{2}}$$

$$= \boldsymbol{\Sigma}_{N_1,N_2}^{-\frac{1}{2}}\left[\left(\mathbb{E}\left(\boldsymbol{\Sigma}_{N_1,N_2}^{\frac{1}{2}}\right)\right) - \boldsymbol{\Sigma}_{N_1,N_2}^{\frac{1}{2}}\right](\mathbb{E}\left(\boldsymbol{\Sigma}_{N_1,N_2}\right))^{\frac{1}{2}} + \boldsymbol{\Sigma}_{N_1,N_2}^{-\frac{1}{2}}\left[(\mathbb{E}\left(\boldsymbol{\Sigma}_{N_1,N_2}\right))^{\frac{1}{2}} - \mathbb{E}\left(\boldsymbol{\Sigma}_{N_1,N_2}^{\frac{1}{2}}\right)\right](\mathbb{E}\left(\boldsymbol{\Sigma}_{N_1,N_2}\right))^{-\frac{1}{2}}$$

*And we have for* (2)*:*

$$(2) = \left(\boldsymbol{\Sigma} + \gamma_{N_1,N_2}\mathbf{I}\right)^{-\frac{1}{2}} \left(\boldsymbol{\Sigma}^{\frac{1}{2}} - \left(\boldsymbol{\Sigma} + \gamma_{N_1,N_2}\mathbf{I}\right)^{\frac{1}{2}}\right) \boldsymbol{\Sigma}^{-\frac{1}{2}}$$

*Thus we have:*

$$\left\| \left(\boldsymbol{\Sigma}_{N_1,N_2}^{-\frac{1}{2}} - \boldsymbol{\Sigma}^{-\frac{1}{2}}\right) \mathbf{S} \right\|_1 \leq \left\| \boldsymbol{\Sigma}_{N_1,N_2}^{-\frac{1}{2}} \left[ \left(\mathbb{E}\left(\boldsymbol{\Sigma}_{N_1,N_2}^{\frac{1}{2}}\right)\right) - \boldsymbol{\Sigma}_{N_1,N_2}^{\frac{1}{2}} \right] (\mathbb{E}\left(\boldsymbol{\Sigma}_{N_1,N_2}\right))^{-\frac{1}{2}} \mathbf{S} \right\|_1$$

$$+ \left\| \boldsymbol{\Sigma}_{N_1,N_2}^{-\frac{1}{2}} \left[ (\mathbb{E}\left(\boldsymbol{\Sigma}_{N_1,N_2}\right))^{\frac{1}{2}} - \mathbb{E}\left(\boldsymbol{\Sigma}_{N_1,N_2}^{\frac{1}{2}}\right) \right] (\mathbb{E}\left(\boldsymbol{\Sigma}_{N_1,N_2}\right))^{-\frac{1}{2}} \mathbf{S} \right\|_1$$

$$+ \left\| \left(\boldsymbol{\Sigma} + \gamma_{N_1,N_2}\mathbf{I}\right)^{-\frac{1}{2}} \left(\boldsymbol{\Sigma}^{\frac{1}{2}} - \left(\boldsymbol{\Sigma} + \gamma_{N_1,N_2}\mathbf{I}\right)^{\frac{1}{2}}\right) \boldsymbol{\Sigma}^{-\frac{1}{2}}\mathbf{S} \right\|_1$$

*But we know that* $|\boldsymbol{\Sigma}_{N_1,N_2}^{-\frac{1}{2}}|_{i,j} \leq \frac{J}{\sqrt{\gamma_{N_1,N_2}}}$ *and by the same reasoning we have also that* $|(\boldsymbol{\Sigma} + \gamma_{N_1,N_2}\mathbf{I})_{i,j}^{-\frac{1}{2}}| \leq \frac{J}{\sqrt{\gamma_{N_1,N_2}}}$. *By noting:*

$$\mathbf{K}_1 = \sup_{k \in [\![1,J]\!]} |[\boldsymbol{\Sigma}^{-\frac{1}{2}}\mathbf{S}]_k|$$

$$\mathbf{K}_2 = \sup_{k \in [\![1,J]\!]} \left| \left[ (\mathbb{E}\left(\boldsymbol{\Sigma}_{N_1,N_2}\right))^{\frac{1}{2}} - \mathbb{E}\left(\boldsymbol{\Sigma}_{N_1,N_2}^{\frac{1}{2}}\right) (\mathbb{E}\left(\boldsymbol{\Sigma}_{N_1,N_2}\right))^{-\frac{1}{2}} \mathbf{S} \right]_k \right|$$

$$\mathbf{K}_3 = \sup_{k \in [\![1,J]\!]} |[(\mathbb{E}\left(\boldsymbol{\Sigma}_{N_1,N_2}\right))^{-\frac{1}{2}} \mathbf{S}]_k|$$

*All these constants are independent from* $N_1, N_2, (x_i)$ *and* $(y_j)$. *We have finally:*

$$\left\| (\boldsymbol{\Sigma}_{N_1,N_2}^{-\frac{1}{2}} - \boldsymbol{\Sigma}^{-\frac{1}{2}})\mathbf{S} \right\|_1 \leq \sum_{j=1}^{J}\sum_{q=1}^{J}\sum_{k=1}^{J} \left| \left(\mathbb{E}\left(\boldsymbol{\Sigma}_{N_1,N_2}^{\frac{1}{2}}\right)\right)_{q,k} - \left(\boldsymbol{\Sigma}_{N_1,N_2}^{\frac{1}{2}}\right)_{q,k} \right| \frac{\mathbf{K}_3 J}{\sqrt{\gamma_{N_1,N_2}}} + J^4\mathbf{K}_1 + \frac{J^3\mathbf{K}_2}{\sqrt{\gamma_{N_1,N_2}}}$$

$$\leq \left[ \sum_{q,k=1}^{J} \left| \left(\mathbb{E}\left(\boldsymbol{\Sigma}_{N_1,N_2}^{\frac{1}{2}}\right)\right)_{q,k} - \left(\boldsymbol{\Sigma}_{N_1,N_2}^{\frac{1}{2}}\right)_{q,k} \right| \right] \frac{\mathbf{K}_3 J^2}{\sqrt{\gamma_{N_1,N_2}}} + J^4\mathbf{K}_1 + \frac{J^3\mathbf{K}_2}{\sqrt{\gamma_{N_1,N_2}}}$$

*And by applying a union bound on all the terms that compose the upper bound of* $|\hat{\lambda}_{N_1,N_2} - \lambda_t|$ *we have thus:*

$$\mathbb{P}\left(\left|\hat{\lambda}_t - \lambda_t\right| \leq \alpha\right) \geq \sum_{k=1}^{J} \mathbb{P}\left(\sqrt{t}\frac{J^2}{\sqrt{\gamma_{N_1,N_2}}} |\mu_X(T_k) - \mu_Y(T_k) - \mathbb{E}(\mu_X(T_k) - \mu_Y(T_k))| \leq \frac{\alpha}{J+J^2}\right)$$

$$+ \sum_{q,k=1}^{J} \mathbb{P}\left(\sqrt{t}\left(\left(\left| \left(\mathbb{E}\left(\boldsymbol{\Sigma}_{N_1,N_2}^{\frac{1}{2}}\right)\right)_{q,k} - \left(\boldsymbol{\Sigma}_{N_1,N_2}^{\frac{1}{2}}\right)_{q,k} \right|\right) \frac{\mathbf{K}_3 J^2}{\sqrt{\gamma_{N_1,N_2}}} + J^4\mathbf{K}_1 + \frac{J^3\mathbf{K}_2}{\sqrt{\gamma_{N_1,N_2}}}\right) \leq \frac{\alpha}{J^2+J}\right)$$

$$- (J^2 + J - 1)$$

*As* $\mu_X(T) - \mu_Y(T) = \sum_{k=1}^{t} Z_i$ *where* $Z_k$ *are independent and:*

- $\forall i \leq N_1, Z_i = \frac{k(x_i,T)}{N_1}$, *so* $|Z_i| \leq \frac{K}{N_1}$

- $\forall N_1 < i \leq N_2, Z_i = -\frac{k(y_i,T)}{N_1}$ *so* $|Z_i| \leq \frac{K}{N_2}$

*We have thanks to Hoeffding's inequality that* $\forall k \in [\![1,J]\!]$ :

$$\mathbb{P}\left(\sqrt{t}\frac{J^2}{\sqrt{\gamma_{N_1,N_2}}} |\mu_X(T_k) - \mu_Y(T_k) - \mathbb{E}(\mu_X(T_k) - \mu_Y(T_k))| \leq \frac{\alpha}{J+J^2}\right)$$

$$\geq 1 - 2\exp\left(-\left(\frac{\alpha}{J^2+J}\right)^2 \frac{\gamma_{N_1,N_2}N_1N_2}{K^2(N_1+N_2)^2}\right)$$

*Moreover $\forall k, q \in [\![1, J]\!]$ :*

$$\mathbb{P}\left[\sqrt{t}\left(\left|\left(\mathbb{E}\left(\mathbf{\Sigma}_{N_1,N_2}^{\frac{1}{2}}\right)\right)_{q,k} - \left(\mathbf{\Sigma}_{N_1,N_2}^{\frac{1}{2}}\right)_{q,k}\right|\frac{\mathbf{K}_3 J^2}{\sqrt{\gamma_{N_1,N_2}}} + \frac{J^4}{\mathbf{K}_1} + \frac{J^3 \mathbf{K}_2}{\sqrt{\gamma_{N_1,N_2}}}\right) \leq \frac{\alpha}{J^2+J}\right]$$

$$= \mathbb{P}\left[\left|\left(\mathbf{\Sigma}_{N_1,N_2}^{\frac{1}{2}}\right)_{k,q} - \mathbb{E}\left(\mathbf{\Sigma}_{N_1,N_2}^{\frac{1}{2}}\right)_{k,q}\right| \leq \frac{\gamma_{N_1,N_2}}{\mathbf{K}_3 J^2}\left[\frac{\alpha}{(J^2+J)\sqrt{t}} - \left(\frac{J^3\mathbf{K}_2}{\sqrt{\gamma_{N_1,N_2}}} + J^4\mathbf{K}_1\right)\right]\right]$$

*Let define $F(x_1, ..., x_{N_1}, y_1, ..., y_{N_2}) := \mathbf{\Sigma}_{N_1,N_2}$ and $F_{k,q}(x_1, ..., x_{N_1}, y_1, ..., y_{N_2}) := (\mathbf{\Sigma}_{N_1,N_2})_{k,q}$ We can see easily that $\forall (x_i), (y_i), x, x', y, y'$:*

$$\left|F_{k,q}(x_1, .., x, .., x_{N_1}, y_1, .., y_{N_2}) - F_{k,q}(x_1, .., x', .., x_{N_1}, y_1, .., y_{N_2})\right| \leq \frac{8}{\rho N_1}$$

*and*

$$\left|F_{k,q}(x_1, .., x_{N_1}, y_1, ..y, .., y_{N_2}) - F_{k,q}(x_1, .., x_{N_1}, y_1, .., y', .., y_{N_2})\right| \leq \frac{8}{(1-\rho)N_2}$$

*Let $g(\mathbf{X}) = \mathbf{X}^{\frac{1}{2}}$ defined on $\mathbf{S}_J^{++}(R)$ and takes values in $\mathbf{S}_J^{++}(R)$. This fuction is well defined because each matrix of $\mathbf{S}_J^{++}(R)$ admits a unique square root matrix on $\mathbf{S}_J^{++}(R)$. Moreover The result hold on $\mathbf{S}_J^{+}(R)$.*

**Lemma 11.** *$g$ is locally Lipschitz continuous on $\mathcal{S}_J^{++}(R)$ which means that:*

$$\forall N > 0, \ \forall \mathbf{X}, \mathbf{Y} \in B(0, N) \subset \mathcal{S}_J^{++}(\mathbb{R}), \quad \exists K_N / \|g(\mathbf{X}) - g(\mathbf{Y})\| \leq K_N \|\mathbf{X} - \mathbf{Y}\|$$

**Proof.** *Let us first prove that $g$ is $C^\infty$. First thanks to Lemma 10 $g$ is continuous on $\mathbf{S}_J^{++}(R)$. Let us show now that $g$ is $C^\infty$ on this space. We know that $\mathbf{M} \to \mathbf{M}^2$ induces a bijection from $\mathcal{S}_n^{++}(R)$ on itself where the inverse is $g$. To prove then that $g$ is $C^\infty$, thanks to the inverse function theorem, we just have to show that $\mathbf{D}_{\mathbf{U}_0}(\mathbf{M} \to \mathbf{M}^2)$ is invertible for every $\mathbf{U}_0 \in \mathcal{S}_n^{++}(R)$. Let $\mathbf{U}_0 \in \mathcal{S}_n^{++}(R)$. And let's consider the differential defined on $\mathcal{S}_n(R)$ in $\mathcal{S}_n(R)$ which is a linear application and which associates $\mathbf{H}$ to $\mathbf{U}_0\mathbf{H} + \mathbf{H}\mathbf{U}_0$. If we prove the injectivity of this function we will have its invertibility as $\mathcal{S}_n(R)$ is a finite dimensional space. Let $\mathbf{H} \in \mathcal{S}_n(R)$ such that $\mathbf{U}_0\mathbf{H} + \mathbf{H}\mathbf{U}_0 = 0$ and $\mathbf{x}$ an eigenvector of $\mathbf{U}_0$ associated with the eigenvalue $\lambda$ which is strictly positive as $\mathbf{U}_0$ is definite positive. We have:*

$$\mathbf{U}_0\mathbf{H}\mathbf{x} = -\mathbf{H}\mathbf{U}_0\mathbf{x} = -\lambda\mathbf{H}\mathbf{x}$$

*As $-\lambda < 0$ it is not an eigenvalue of $\mathbf{U}_0$ and then $\mathbf{H}\mathbf{x} = 0$. This is true for all the eigenvectors of $\mathbf{U}_0$, then $\mathbf{H} = 0$ and the differential is injective, so $g$ is $C^\infty$ on $\mathcal{S}_n^{++}(R)$. Finally by applying the Mean value theorem, we have that $g$ is locally Lipschitz continuous.*

*We also remark that $\|F(x_i, y_j)\| = \max_{i,j\in[\![1,J]\!]} |(\mathbf{\Sigma}_{N_1,N_2})_{i,j}| \leq \lambda$ (because the Gaussian kernel is bounded) with $\lambda$ independent from $N_1, N_2, (x_i)$ and $(y_j)$. Then by taking the following norm $\|\mathbf{M}\| = \max_{i,j\in[\![1,J]\!]} \mathbf{M}_{i,j}$ we have:*

$$\left\|g(F(x_1, .., x, .., x_{N_1}, y_1, .., y_{N_2})) - g(F(x_1, .., x', .., x_{N_1}, y_1, .., y_{N_2}))\right\|$$

$$\leq K_\lambda \left\|F(x_1, .., x, .., x_{N_1}, y_1, .., y_{N_2}) - F(x_1, .., x', .., x_{N_1}, y_1, .., y_{N_2})\right\|$$

*And:*

$$\left\|F(x_1, .., x, .., x_{N_1}, y_1, .., y_{N_2}) - F(x_1, .., x', .., x_{N_1}, y_1, .., y_{N_2})\right\| \leq \max\left(\frac{8}{\rho N_1}, \frac{8}{(1-\rho)N_2}\right)$$

*Then $\forall k, q \in [\![1, J]\!]$ :*

$$\left|\mathbf{\Sigma}_{N_1,N_2}^{\frac{1}{2}}(x) - \mathbf{\Sigma}_{N_1,N_2}^{\frac{1}{2}}(x')\right| \leq K_\lambda \max\left(\frac{8}{\rho N_1}, \frac{8}{(1-\rho)N_2}\right)$$

*And thanks to the McDiarmid inequality we have:*

$$\mathbb{P}\left(\left|\left(\mathbf{\Sigma}_{N_1,N_2}^{\frac{1}{2}}\right)_{k,q} - \mathbb{E}\left(\mathbf{\Sigma}_{N_1,N_2}^{\frac{1}{2}}\right)_{k,q}\right| \leq \frac{\gamma_{N_1,N_2}}{K_3 J^2}\left[\frac{\alpha}{(J^2+J)\sqrt{t}} - \left(\frac{J^3 K_2}{\sqrt{\gamma_{N_1,N_2}}} + J^4 K_1\right)\right]\right)$$

$$\geq 1 - 2\exp\left(-2\frac{\left(\frac{\gamma_{N_1,N_2}}{K_3 J^2}\left(\frac{\alpha}{(J^2+J)\sqrt{t}} - \frac{J^3 K_2}{\sqrt{\gamma_{N_1,N_2}}} - J^4 K_1\right)\right)^2}{K_\lambda^2(N_1+N_2)\max\left(\frac{8}{\rho N_1},\frac{8}{(1-\rho)N_2}\right)^2}\right)$$

*Then we have:*

$$\mathbb{P}\left(|\widehat{\lambda}_{N_1,N_2} - \lambda_t| \leq \alpha\right) \geq 1 - 2\sum_{k=1}^{J}\exp\left(-\left(\frac{\alpha}{J^2+J}\right)^2\frac{\gamma_{N_1,N_2}N_1 N_2}{(N_1+N_2)^2}\right)$$

$$- 2\sum_{k,q=1}^{J}\exp\left(-2\frac{\left(\frac{\gamma_{N_1,N_2}}{K_3 J^2}\left(\frac{\alpha}{(J^2+J)\sqrt{t}} - \frac{J^3 K_2}{\sqrt{\gamma_{N_1,N_2}}} - J^4 K_1\right)\right)^2}{K_\lambda^2(N_1+N_2)\max\left(\frac{8}{\rho N_1},\frac{8}{(1-\rho)N_2}\right)^2}\right)$$

*And finally, by taking $\alpha = \lambda_t - \delta$ we have the result.*

## E   Using smooth characteristic functions (SCF)

**Theorem 5.** *Let $k$ be an analytic, integrable kernel with an inverse Fourier transform strictly greater than zero. For any $J > 0$, we define:*

$$d_{\Phi,J} = \left\{d_{\Phi,J}[p,q] = \frac{1}{J}\sum_{j=1}^{J}|\Phi_p(T_j) - \Phi_q(T_j)|: \quad p,q \in \mathcal{M}_+^1\left(\mathbb{R}^d\right), \Phi_p, \Phi_q \in L_1\left(\mathbb{R}^d\right)\right\}$$

*Then for any $J > 0$, $d_{\Phi,J}$ is a random metric on the space of Borel probability measures with integrable characteristic functions.*

**Proof.** *Since $k$ is an analytic, integrable kernel with an inverse Fourier transform strictly then zero then by the Lemma 3 the mapping $\Lambda : P \to \Phi_P$ is injective and $\Lambda(P)$ is an element of the RKHS associated with $k$. The Lemma 7 shows that $\Phi_P$ is analytic. Therefore we can use Lemma 6 to see that $d_{\Lambda,J}(P,Q) = d_{\Phi,J}(P,Q)$ is a random metric. This concludes the proof of the Theorem.*

### E.1   Proof of Proposition 3.4

**Proposition 5.** *Let $\alpha \in ]0,1[$, $\gamma > 0$ and $J \geq 2$. Let $\{T_j\}_{j=1}^{J}$ sampled i.i.d. from the distribution $\Gamma$ and let $X := \{x_i\}_{i=1}^{n}$ and $Y := \{y_i\}_{i=1}^{n}$ i.i.d. samples from $P$ and $Q$ respectively. Let us denote $\delta$ the $(1-\alpha)$-quantile of the asymptotic null distribution of $\widehat{d}_{\ell_1,\Phi,J}[X,Y]$ and $\beta$ the $(1-\alpha)$-quantile of the asymptotic null distribution of $\widehat{d}_{\ell_2,\Phi,J}^2[X,Y]$. Under the alternative hypothesis, almost surely, there exists $N \geq 1$ such that for all $n \geq N$, with a probability of at least $1 - \gamma$ we have:*

$$\widehat{d}_{\ell_2,\Phi,J}^2[X,Y] > \beta \Rightarrow \widehat{d}_{\ell_1,\Phi,J}[X,Y] > \delta \tag{29}$$

**Proof.** *Let us first introduce the following Lemma:*

**Lemma 12.** *Let $\mathbf{x}$ a random vector $\in \mathbb{C}^J$ with $J \geq 2$, $\epsilon > 0$, $\gamma > 0$ and $\mathbf{z} := \min_{j\in[|1,J|]}|Re(x_j)| + |Im(x_j)|$ where $Im$ and $Re$ are respectively the imaginary and real part functions. Moreover let denote $\mathbf{X} := (Im(x_j), Re(x_j))_{j=1}^{J} \in \mathbb{R}^{2J}$. If*

$$\mathbb{P}(\mathbf{z} \geq \epsilon) \geq 1 - \gamma$$

*we have with a probability of at least $1 - \gamma$ that, $\forall t_1 \geq t_2 \geq 0$, if $\epsilon \geq \sqrt{\frac{t_1^2 - t_2^2}{J(J-1)}}$, then*

$$\|\mathbf{X}\|_2 > t_2 \Rightarrow \|\mathbf{X}\|_1 \geq t_1.$$

**Proof.** *First we remarks that:*

$$\epsilon > \sqrt{\frac{t_1^2 - t_2^2}{J(J-1)}} \Rightarrow J(J-1)\,\epsilon > t_1^2 - t_2^2$$

$$\Rightarrow t_2^2 > t_1^2 - J(J-1)\,\epsilon^2$$

*Therefore, we have:*

$$\|\mathbf{X}\|_2 \geq t_2 \Rightarrow \|\mathbf{X}\|_2^2 + J(J-1)\,\epsilon^2 \geq t_1^2$$

$$\Rightarrow \sqrt{\|\mathbf{X}\|_2^2 + J(J-1)\,\epsilon^2} \geq t_1$$

*But we have that:*

$$\|\mathbf{X}\|_1^2 = \|\mathbf{X}\|_2^2 + \sum_{i \neq j} (|Im(x_i)| + |Re(x_i)|)(|Im(x_j)| + |Re(x_j)|)$$

*Therefore we have with a probability of 1-$\gamma$ that:*

$$\|\mathbf{X}\|_1^2 \geq \|\mathbf{X}\|_2^2 + J(J-1)\,\epsilon^2$$

*And:*

$$\|\mathbf{X}\|_2 \geq t_2 \Rightarrow \|\mathbf{X}\|_1 \geq t_1$$

*Moreover by denoting $\delta$ the $(1-\alpha)$-quantile of the asymptotic null distribution of $\widehat{d}_{\ell_1,\Phi,J}[X,Y]$ and $\beta$ the $(1-\alpha)$-quantile of the asymptotic null distribution of $\widehat{d}_{\ell_2,\Phi,J}^2[X,Y]$ thanks to Lemma 9, we have that $\delta \geq \sqrt{\beta}$. Therefore to show the result we only need to show that the assumption of the Lemma 12 is sastified for the random vector $\mathbf{X} := \sqrt{n}\mathbf{S}_n \in \mathbb{R}^{2J}$, $t_1 = \delta$ and $t_2 = \sqrt{\beta}$, i.e. for $\epsilon = \sqrt{\frac{\delta^2 - \beta}{J(J-1)}}$ under the alternative hypothesis. Under $H_1 : P \neq Q$, we have $\mathbf{S}_n$ converges in probability to the vector $\mathbf{S}$ where $\boldsymbol{\Sigma} := \mathbb{E}_{(x,y)\sim(p,q)}(\Sigma_n)$ and $\mathbf{S} := \mathbb{E}_{(x,y)\sim(p,q)}(\mathbf{S}_n)$. Moreover we have $\mathbf{S} = (Im(\Phi_P(T_j) - \Phi_Q(T_j)), Re(\Phi_P(T_j) - \Phi_Q(T_j)))_{j=1}^J \in \mathbb{R}^{2J}$. Indeed, according to the Definition 2, we have for all $j \in [|1,J|]$:*

$$\phi_P(T_j) := \int_{\epsilon \in \mathbb{R}^d} \psi_P(\epsilon) k(T_j - \epsilon) d\epsilon$$

$$= \int_{\epsilon \in \mathbb{R}^d} \int_{x \in \mathbb{R}^d} \exp(ix^T \epsilon) k(T_j - \epsilon) dP(x) d\epsilon$$

$$= \int_{x \in \mathbb{R}^d} \left( \int_{\epsilon \in \mathbb{R}^d} \exp(ix^T(\epsilon - T_j)) k(\epsilon - T_j) \right) \exp(ix^T T_j) d\epsilon dP(x)$$

$$= \int_{x \in \mathbb{R}^d} f(x) \exp(ix^T T_j) dP(x)$$

*and all these equalities hold as $k$ is integrable. Lemma 3 guarantees the injectivity of the function $\Gamma : P \to \Phi_P$, and as $P \neq Q$, therefore $\Phi_P - \Phi_Q$ is a non-zero function. Moreover $\Phi_P$ and $\Phi_Q$ live in the RKHS $H_k$ associated with $k$. Therefore thanks to Lemma 7, $\Phi_P - \Phi_Q$ is analytic. Therefore thanks to Lemma 5, $\Phi_P - \Phi_Q$ is almost surely non zero. Moreover the $(T_j)_{j=1}^J$ are independent, therefore almost surely $(|\Phi_P(T_j) - \Phi_Q(T_j)|)_j$ are all non zero, and then $(|Im(\Phi_P(T_j) - \Phi_Q(T_j))| + |Re(\Phi_P(T_j) - \Phi_Q(T_j))|)_j$ are all non zero. Then by continuity of the functions defined for all $k \in [|1,J|]$ by:*

$$\phi_k : x := (x_j^1, x_j^2)_{j=1}^J \in \mathbb{R}^{2J} \to |x_k^1| + |x_k^1| \tag{30}$$

*We have that for all $k \in [|1,J|]$, $\phi_k(\mathbf{S}_n)$ converge in probability towards $\phi_k(\mathbf{S})$, which are almost surely all non zeros. Then for all $k \in [\![1,J]\!]$ we have:*

$$\mathbb{P}\left(\left|(\sqrt{n}\phi_k(\mathbf{S}_n)\right| > \epsilon\right) = \mathbb{P}_{X,Y}\left(\left|(\phi_k(\mathbf{S}_n)\right| - \frac{\epsilon}{\sqrt{n}} > 0\right)$$

*And as $\frac{\epsilon}{\sqrt{n}} \to 0$ as $n \to \infty$, we have finally almost surely for all $k \in [\![1,J]\!]$:*

$$\mathbb{P}_{X,Y}\left(\left|(\sqrt{n}\phi_k(\mathbf{S}_n)\right| \geq \epsilon\right) \to 1 \text{ as } n \to \infty$$

*Therefore almost surely there exist $N \geq 1$ such that for all $n \geq N$ and for all $k \in [\![1, J]\!]$:*

$$\mathbb{P}_{X,Y} \left( \left| (\sqrt{n} \phi_k(\mathbf{S}_n) \right| \geq \epsilon \right) \geq 1 - \frac{\gamma}{J}$$

*Finally by applying a union bound we obtain that almost surely, for all $n \geq N$:*

$$\mathbb{P}_{X,Y} \left( \forall k \in [|1, J|], \left| (\sqrt{n} \phi_k(\mathbf{S}_n) \right| \geq \epsilon \right) \geq 1 - \gamma$$

*Therefore by applying Lemma 8, we obtain that, almost surely, for all $n \geq N$, with a probability of at least $1 - \gamma$:*

$$\widehat{d}_{\ell_2, \Phi, J}[X, Y] > \sqrt{\beta} \Rightarrow \widehat{d}_{\ell_1, \Phi, J}[X, Y] > \delta$$

# F  Experiments

## F.1  Realization of the $\ell_1$-based tests

Indeed to realize these tests, we need to compute the $1 - \alpha$ quantile of the $Nake\left(\frac{1}{2}, 1, J\right)$. To do so we need to obtain the cumulative distribution function ( CDF ) of the sum of $J$ Nakagami i.i.d. But as we do not have a closed form of this distribution, we need to estimate this CDF by considering the empirical distribution function. Indeed to generate samples from $Nake\left(\frac{1}{2}, 1, J\right)$, it is sufficient to generate samples from multivariate normal distribution $\mathcal{N}\left(0, I_{d_J}\right)$, and to sum the absolute values of the $J$ coordinates of theses vectors.

Moreover, we have the following result:

**Theorem 6.** *(Dvoretzky–Kiefer–Wolfowitz inequality) Let $x_1,...,x_n$ be real-valued independent and identically distributed random variables with cumulative distribution function $F(.)$ Let $F_n$ denote the associated empirical distribution function defined by:*

$$F_n(t) = \frac{1}{n} \sum_{i=1}^{n} \mathbf{1}_{x_i \leq t}$$

*Then we have $\forall \epsilon > 0$:*

$$\mathbb{P}\left( ||F_n - F||_\infty > \epsilon \right) \leq 2e^{-2n\epsilon^2}$$

Finally we have, $F(x) - \epsilon \leq F_n(x) \leq F(x) + \epsilon$ with a probability of $1 - \delta$ where $\epsilon = \sqrt{\frac{ln\left(\frac{2}{\delta}\right)}{2n}}$.

Then with a probability of 99%, and by taking $n = 100\,000$ samples i.i.d of the $Naka\left(\frac{1}{2}, 1, J\right)$, we can estimate the CDF with an error of $\epsilon \leq 0.0051$, which is less than $\alpha = 0.01$.

**Optimization:** The lower bounds that we optimize to perform **L1-opt-ME** and **L1-opt-SCF** are non-convex, as in the prior art [17]. However, the use of the $\ell_1$-norm makes optimization even harder, as it is no longer a smooth. Moreover we need to differentiate through the inverse square root matrix operation which can lead is some cases to degenerate matrices during the gradient ascent. Therefore to avoid this, we decide to check at each step the convergence of the inverse square root matrix operation. Further work should consider dedicated optimization algorithms.

Table 3 gives the run times of the different optimized tests on the Blobs problem when the test sample size is $n^{te} = 1e6$.

|  | L1-opt-ME | ME-full | L1-opt-SCF | SCF-full |
|---|---|---|---|---|
| Run Time (s) | 164.23 | 157.97 | 599.77 | 579.42 |

Table 3: Run times of the optimized tests when $n^{te} = 1e6$ and $J = 2$ for the blobs problem.

**Software implementation**: as the expression of the optimization objective is rather complicated, we use the automatic differentiation of pytorch [23], to compute its gradient, and then proceed with a gradient ascent where the step size after $t$ iterations is the inverse of the euclidean norm of the gradient times $\sqrt{t}$. The specific code can be found at `https://github.com/meyerscetbon/l1_two_sample_test`.

## F.2 Experimental verification of the Propostion 3.1

To show the validity of the Proposition 3.1 experimentally, we examine the behavior of the unormalized $\ell_2$ and $\ell_1$ based tests respectively defined in eq. 6 and 8. In Figure 5, we compare the unormalized tests on the GVD problem where we increase the test sample size with $d = 100$ and $J = 2$. Here the locations are chosen at random and are sampled from a standard normal distribution. Moreover here $\alpha = 0.05$. Compared to the normalized tests studied in section 4 is that here we no longer have a direct access to the quantiles of the asymptotic null distribution. Being a problem where we can generated the data ourselves, we have therefore estimate the quantiles of our interest. Moreover, when comparing the $\ell_2$ and $\ell_1$ tests, we sample at random the locations and we evaluate the two statistics at the same locations. We see that as the test sample size increases, the $\ell_1$-based tests rejects better the null distribution.

Figure 5: Plot of type-II error against the test sample size $n^{te}$ in the GVD toy problem: $P = \mathcal{N}(0, I_d)$ and $Q = \mathcal{N}(0, \text{diag}(2, 1, ..., 1))$ with $d = 100$.

## F.3 Experiments on a more difficult problem

Figure 6: Plot of type-II error against the test sample size $n^{te}$ in the following toy problem: $P = \mathcal{N}(0, I_d)$ and $Q = \mathcal{N}\left((0.3, 0, .., 0)^T, I_d\right)$ with $d = 100$

In Figure 6, we consider the following GMD problem: $P \sim \mathcal{N}(0, I_d), Q \sim \mathcal{N}\left((0.3, 0, .., 0)^T, I_d\right)$ with $d = 100$. The figure shows that when the problem of GMD is more difficult, we can see that **L1-opt-ME** performs the best.

## F.4 Informative features

We show that the optimization of the proxy $\widehat{\lambda}_t^{tr}(\theta)$ for the test power in the $\ell_1$ case is informative for revealing the difference of the two samples in the ME test as in [17] with the $\ell_2$ version. We consider the Gaussian Mean Difference (GMD) problem (see Table 1), where both $P$ and $Q$ are two-dimensional normal distributions with different means. We use $J = 2$ test locations $T_1$ and $T_2$, where $T_1$ is fixed to the location indicated by the black triangle in Figure 7. The contour plot shows $T_2 \to \widehat{\lambda}_t^{tr}(T_1, T_2)$.

Figure 7a suggests that $\widehat{\lambda}_t^{tr}(T_1, T_2)$ is maximized when $T_2$ is placed in either of the two regions that captures the difference of the two samples i.e., the region in which the probability masses of $P$ and $Q$ have less overlap. In Figure 7b, we consider placing $T_1$ in one of the two key regions. In this case, the contour plot shows that $T_2$ should be placed in the other region to maximize $\widehat{\lambda}_t^{tr}(T_1, T_2)$, implying that placing multiple test locations in the same neighborhood does not increase the discriminability. The two modes on the left and right suggest two ways to place the test location in a region that reveals the difference. The non-convexity of the $\widehat{\lambda}_t^{tr}(T_1, T_2)$ is an indication of many informative ways to detect differences of $P$ and $Q$, rather than a drawback. A convex objective would not capture this multimodality.

Figure 7: **Illustrating interpretable features**, replicating in the $\ell_1$ case the figure of [17]. A contour plot of $\widehat{\lambda}_t^{tr}(T_1, T_2)$ as a function of $T_2$, when $J = 2$, and $T_1$ is fixed. The red and black dots represent the samples from the $P$ and $Q$ distributions, and the big black triangle the position of $T_1$.

(a) $T_1$ is centered

(b) $T_1$ lives in the left region

| P | Q | L1-opt-ME | L1-grid-ME | L1-opt-SCF | L1-grid-SCF | ME-full | SCF-full |
|---|---|---|---|---|---|---|---|
| sci (1187) | sci (1187) | **0.00** | **0.00** | 0.004 | **0.00** | 1 | 0.002 |
| sci (1187) | comp (292) | **0.00** | 0.496 | **0.00** | 0.170 | **0.00** | 0.634 |
| sci (1187) | alt (240) | **0.00** | 0.370 | **0.00** | 0.064 | **0.00** | 0.510 |

Table 4: Type-I errors and Type-II errors of various the L1-tests in the problem of distinguishing the newsgroups text dataset. $\alpha = 0.01$. $J = 2$. The number in brackets denotes the test sample size of each samples.

## F.5 Real problem: 20 newsgroups text dataset

In this experiment we use the 20 newsgroups text dataset from [18] which comprises around 18000 newsgroups posts on 20 topics. We consider 3 categories which are: "comp", "sci", and "alt" . The first category is about components in hardware systems, the second is about sciences and spaces, and the last is about religion. To perform the tests we need to embed these documents in a metric space. For this, we use the TF-IDF matrix by group of two categories with a $df \geq 30$, which lead to embed the documents in spaces of 3 000 dimensions approximately. Then we perform the two-sample tests on the embedded documents. We compare the distribution of "sci" documents with others, as well as with itself to evaluate the level of the tests. The number of samples of each category is not the same, hence to perform the tests from [17], we take randomly $n_{min}$ samples for both distributions without replacement (where $n_{min}$ is in fact the number of samples of the distributions compared to the sci distribution). We set the number of location $J = 2$.

Type-I errors and type-II errors are summarized in Table 4 The two first columns indicates the categories of the papers in the two samples. This task represents a case in which $H_0$ holds. In this case all the tests are conservative except the **ME-full** test which totally rejecting the null hypothesis. In the other problems, we show the Type-II errors of our tests. The $\ell_1$ optimized tests perform very well, which shows that the locations learned are indeed discriminant. The $\ell_1$ approaches bring a clear gain in statistical control compared to their $\ell_2$ counterparts.

## F.6 Real problem: fast-food distribution

| Problem | L1-opt-ME | L1-grid-ME | L1-opt-SCF | L1-grid-SCF | ME-full | SCF-full | MMD-quad |
|---|---|---|---|---|---|---|---|
| McDo vs McDo (2002) | 0.010 | 0.000 | 0.000 | 0.000 | 0.012 | 0.000 | 0.000 |

Table 5: **Fast food dataset:** Type-I errors for distinguishing the distribution of fast food restaurants. $\alpha = 0.01$. $J = 3$. The number in brackets denotes the sample size of the distribution on the right. We consider MMD-quad as the gold standard.

Table 5 summarizes Type-I errors observed on the Mac Donald's vs Mac Donald's problem. It shows that the optimized tests based on mean embeddings stay roughly at the specified level $\alpha = 0.01$ when $H_0$ hold, and others are more conservative.

Figures 9, 10, 11, 12, 13 give the distributions of the data (restaurant locations) and of the $T_J$ for each of the problems that we consider.

Figure 8: **Fast food data:**
Visualizing interpretable locations for differences in Mc Donald's vs Burger King and Mc Donald's vs Wendy's. The lines correspond to the distribution of the locations chosen for the $T_J$ features by the L1-opt-ME procedure. The distributions are estimated with a kernel density estimate. The lines represent the contours probabilities 80% and 90%.

Figure 9: **Mc Donald's vs Burger King**

Figure 10: **Mc Donald's vs Taco Bell**

Figure 11: **Mc Donald's vs Wendy's**

Figure 12: **Mc Donald's vs Arby's**

Figure 13: **Mc Donald's vs KFC**