[Reviews · NeurIPS 2019]

Reviewer 1



The paper is well written. The authors provide a generalisation of the approach taken in [5] which seems quite natural. Their approximation of the metric seems also quite inspired by the approximation performed in [5]. I think the paper is complete, the methodological results are well justified and the experimental study is extensive. I think this paper "Interpoint Distance Based Two Sample Tests in High Dimension" is relevant as they show a very similar behaviour of L1-based test in higher dimensions Typos: 1) In line 73, I think D_{\mu,J} has not been previously defined. --------------------------- Post review comments: I thank the authors for their clear response. I think this paper is a good submission so I would like to maintain my overall score.

Reviewer 2



Summary: The paper showed that two samples testing statistics between kernel based distribution representatives (i.e. kernel mean embeddings and smoothed characteristic functions) using L^2 distance [5] can be generalised to any L^p distance with p>=1. Theorem 2.1 and Theorem 3 of the paper showed that this definition give rise to a metric on the space of Borel probability measures and that it metrise the weak convergence. The paper showed when using L^1 distance instead of L^2 distance, the power of the tests of [5] are better with higher probabilities [Proposition 3.1 and 3.4]. Like [5], the paper considered the asymptotic null distribution of the normalised difference between distributions representatives, which give close form asymptotic null distributions [Proposition 3.2]. Further, they provided lower bound on the test power of these two l1-based tests [Proposition 3.3]. This results led to the conclusion that it is sufficient to optimise the test statistics jointly in the kernel parameter and the test locations in order to maximise such lower bound. Empirically, the paper investigated the proposed methods on 4 synthetic and 3 real data problems, illustrating the benefits of using the proposed l1 geometry. ====== Clarity: Overall, the paper is well written with clear logical structure. Notations are clearly define except at a few small places that it was mentioned before definition (e.g. line 73: D_\mu, J was not defined?) I understand that the authors follow the synthetic experimental set up of [16] and therefore chosen the same parameter settings. It would be nice to inform the readers again of these numbers. Also in line 221, the blobs experiments with dim = 30, is that a typo? Quality: The paper presented several theoretical results with proofs on statistics with l1 norm. In the case of two samples testing, they have shown theoretically that l1 geometry gives better statistical power. Such claim is also well supported by experimental results on real and synthetic datasets. Though, I hope to see a little more intuitions on the theoretical results presented. I am wondering if the authors could comment on any weaknesses of their work? Do we lose anything when change the norm from l2 to l1? Or are the authors trying to say that practitioners should now always use l1 over l2 norm? It seems when the samples sizes are large, the run time of the optimised proposed test is longer than the other linear time tests (though still small in this case). Have the authors tried on problems that require even large samples (e.g. 10^7)? For a given computation time, should one opt for the l1-based tests over a l2-based test? Originality & Significance: The methods proposed are a clear significant extension/ generalisation of the work from [5]. The theoretical results are important. The l1-norm based two samples tests proposed seems likely to be used widely in various applications competing with/advancing performance of the current state of art. ===================== Authors feedback read. I am happy with the response provided by the authors.

Reviewer 3



I thank the authors for their response, and would like to maintain my overall evaluation. ===== The paper is generally well-written, and addresses an important problem. In the literature on kernel two-sample tests, apart from the quadratic-time maximum mean discrepancy (MMD) test which computes the RKHS norm of the mean embeddings, [5] examined a test statistic based on the L^2 distance of the empirical mean embeddings, and gave the asymptotic null distribution in terms of Chi-squared variables. The current work generalizes this approach by replacing the L^2 distance with general L^p distances, and prove that for p >= 1, the resulting metrics dominate weak convergence. For the test based on the L1 norm, the paper further derives the asymptotic null distribution in terms of Nakagami variables, and proves that it achieves lower type-II error than the test based on the L2 norm. Overall, I feel that the paper makes an interesting observation and a valuable contribution. More specifically, I have the following questions/comments: - How does one choose the distribution \Gamma from which the T_j's are sampled? - It would be helpful to clearly state the computational complexity of the proposed tests in terms of N1, N2 and J. - In the experiments, to fully verify Proposition 3.1, I think it would be helpful to have a comparison of the Type-I and Type-II errors of all the tests without optimizing for test locations, since the latter introduce an additional source of confounding. - In the second panel of Figures 1 and 2, why are the ME lines missing? - Table 2 is rather confusing as I'm not sure which entries represent Type-I errors and which represent Type-II errors. It might be better to create two separate tables and defer e.g., the Type-I error one to the supplementary material. - Since the Nakagami distribution may not be familiar to most readers, it would be helpful to provide its pdf in the main text or supplementary material to avoid confusion regarding its parameterization. - I would recommend the authors to release code for implementing the tests and for reproducing the experiments.

[Author Response · NeurIPS 2019]

We thank the reviewers for their thorough reading of our work.

We thanks Rev 1 for pointing out the results of the preprint [3]. Indeed [3] study a class of two sample test statistics based on inter-point distances and they show benefits of using the $\ell_1$ norm over the Euclidean distance and the Maximum Mean Discrepancy (MMD) when the dimensionality goes to infinity. For this class of test statistics, they characterize asymptotic power loss w.r.t the dimension and show that the $\ell_1$ norm is beneficial compared to the $\ell_2$ norm and the MMD provided that the summation of discrepancies between marginal univariate distributions is large enough. These results echo our message that $\ell_1$-based tests outperform their $\ell_2$ counterparts and we will mention them in the final version. Our work is complementary and differs on several aspects. The population version of the $\ell_1$-based statistic ($\mathbf{ED}^1$) proposed in [3] does not fully characterize the difference between two distributions $P, Q$ since $\mathbf{ED}^1(P, Q) = 0$ does not necessarily imply $P = Q$ while the population version of our $\ell_1$-based tests are metric on the space of Borel probability measures which metrize the weak convergence. Our experiment varying the dimensionality of the problem is the part of our work that is closest to [3], and confirms their general message, with a different test procedure. However, the test statistics introduced in [3] have a quadratic cost in the sample size and the tests realized are permutation based tests, which can be very expensive when the sample size increases while our tests are linear in the sample size. With regards to the link to [1], our work adapts the idea of deriving (Monte Carlo) approximation to our $\ell_1$ metric, while [1] had studied a related approximation in the Euclidean case. It also establishes many more results on the $\ell_1$-based metric, including the IPM formulation, the weak convergence properties, and the lower bound on the test power. Moreover we show both theoretically and empirically the benefits of using the $\ell_1$ norm compared to the Euclidean geometry.

Following the suggestion of the Rev 2, we will explain the exact parameter settings, which is the same as in [2], that we used for the initialization of the test locations, the Gaussian width and the value used for regularization parameter $\gamma_{N_1,N_2}$ to compute the optimized tests **L1-opt-ME** and **L1-opt-SCF**. The statistics presented in our paper capture differences between distribution representatives in a RKHS at J locations. When using analytic kernels, these differences between representatives become dense. Therefore when the sample size is large enough, these dense differences are large enough to allow the $\ell_1$-norm to reject better the null hypothesis with high probability. The most important weakness of our study is probably the optimization of the cost functions. The lower bound that we optimize is non-convex, as in the $\ell_2$ prior art [2]. However, the use of the $\ell_1$-norm makes optimization even harder, as it is no longer a smooth. Further work should consider dedicated optimization algorithms. We will try to add a mention of this weakness in the manuscript. In Table 1, we run the different optimized tests on the Blobs problem when the test sample size is $n^{te} = 1e6$.

Table 1: Higgs dataset: Table of the running times of the optimized tests when $n^{te} = 1e6$ and $J = 2$.

|  | **L1-opt-ME** | **ME-full** | **L1-opt-SCF** | **SCF-full** |
|---|---|---|---|---|
| Running Time (s) | 164.23 | 157.97 | 599.77 | 579.42 |

To answer Rev 3, in practice, the distribution $\Gamma$ that we use to sample the $\{T_j\}_{j=1}^J$ to compute the grid version of our statistics, **L1-grid-ME** and **L1-grid-SCF**, is a multivariate normal distribution (see line 197). More specifically, for both tests, we sample the test locations with realizations from two multivariate normal distributions fitted to samples from P and Q; this ensures that the locations are well supported by the data. By denoting $t = N_1 + N_2$, for both tests the testing cost is $\mathcal{O}(J^3 + Jt + dJt)$ and the optimization costs is $\mathcal{O}(J^3 + dJt)$ per gradient ascent (see line 214 - 215). In the second panel of Figures 1 and 2, the ME lines are drawn but their reach the minimum Type-II error of 0.0 when the dimension is between 5 and 1500 or the sample test size is between 500 and 5500. Indeed the GMD problem is an easy problem for this test. Therefore we decide to plot in the supp. mat. an harder version of this problem where the difference between means is smaller (see Figure 4). In Table 2, the McDo vs McDo problem corresponds to the Type-I errors and the other problems represent Type-II errors. We will follow the suggestion proposed by Rev 3 and create two separate tables for the Type-I errors and Type-II error respectively. We will also add the pdf of the Nakagami distribution in the main text. To show experimentally the results of the Proposition 3.1 and 3.4, we need to compute the quantiles of the asymptotic null distributions of the different unnormalized tests that we have presented which require a computationally-costly bootstrap or permutation procedure. As we do not have computational ressources available before the end of the author response period to run a full experiment, we prefer to provide such results later but they will be added in the final manuscript. We will also try to release the code as early as possible.

# References

[1] K. P. Chwialkowski, A. Ramdas, D. Sejdinovic, and A. Gretton. Fast two-sample testing with analytic representations of probability measures. In *Advances in Neural Information Processing Systems*, page 1981, 2015.

[2] W. Jitkrittum, Z. Szabó, K. P. Chwialkowski, and A. Gretton. Interpretable distribution features with maximum testing power. In *Advances in Neural Information Processing Systems*, pages 181–189, 2016.

[3] C. Zhu and X. Shao. Interpoint distance based two sample tests in high dimension. *arXiv:1902.07279*, 2019.


[Meta-Review · NeurIPS 2019]

The paper shows L_p distance of kernel mean embedding as a distance measure of distributions, and shows that L_1-based statistics has higher power than the standard L2 distance. The paper is clearly written and technically sound. The proposed method will be beneficial in the field of kernel-based distance measures and statistical test.